# Organellar Genomes of *Sargassum hemiphyllum* var. *chinense* Provide Insight into the Characteristics of Phaeophyceae

**DOI:** 10.3390/ijms25168584

**Published:** 2024-08-06

**Authors:** Xuli Jia, Weizhou Chen, Tao Liu, Zepan Chen

**Affiliations:** 1College of Marine Life Sciences, Ocean University of China, Qingdao 266003, China; 18734539035@163.com; 2Marine Biology Institute, Shantou University, Shantou 515063, China; wzchen@stu.edu.cn (W.C.); zepan@stu.edu.cn (Z.C.); 3State Key Laboratory of Marine Environmental Science, College of Ocean and Earth Science, Xiamen University, Xiamen 361102, China

**Keywords:** *Sargassum hemiphyllum* var. *chinense*, Phaeophyceae, mitochondrial DNA (mtDNA), chloroplast DNA (cpDNA), comparative genomics, phylogenetic analysis

## Abstract

*Sargassum hemiphyllum* var. *chinense*, a prevalent seaweed along the Chinese coast, has economic and ecological significance. However, systematic positions within *Sargassum* and among the three orders of Phaeophyceae, Fucales, Ectocarpales, and Laminariales are in debate. Here, we reported the organellar genomes of *S. hemiphyllum* var. *chinense* (34,686-bp mitogenome with 65 genes and 124,323 bp plastome with 173 genes) and the investigation of comparative genomics and systematics of 37 mitogenomes and 22 plastomes of Fucales (including *S. hemiphyllum* var. *chinense*), Ectocarpales, and Laminariales in Phaeophyceae. Whole genome collinearity analysis showed gene number, type, and arrangement were consistent in organellar genomes of *Sargassum* with 360 SNP loci identified as *S. hemiphyllum* var. *chinense* and two genes (*rps*7 and *cox*2) identified as intrageneric classifications of *Sargassum*. Comparative genomics of the three orders of Phaeophyceae exhibited the same content and different types (*pet*L was only found in plastomes of the order Fucales and Ectocarpales) and arrangements (most plastomes were rearranged, but *trn*A and *trn*D in the mitogenome represented different orders) in genes. We quantified the frequency of RNA-editing (canonical C-to-U) in both organellar genomes; the proportion of edited sites corresponded to 0.02% of the plastome and 0.23% of the mitogenome (in reference to the total genome) of *S. hemiphyllum* var. *chinense*. The repetition ratio of Fucales was relatively low, with scattered and tandem repeats (nine tandem repeats of 14–24 bp) dominating, while most protein-coding genes underwent negative selection (Ka/Ks < 1). Collectively, these findings provide valuable insights to guide future species identification and evolutionary status of three important Phaeophyceae order species.

## 1. Introduction

*Sargassum* spp. are economically important macroalgal species, representing one of the most abundant brown algal (Ochrophyta, Phaeophyceae, Fucales, and Sargassaceae) genera, being widely distributed in temperate and tropical oceans [1]. This genus has broad application prospects as well as important roles in aquaculture, nutrition, and climate change mitigation, in addition to serving as environmental indicators for the establishment and stability of aquatic ecosystems [2,3,4,5,6,7,8]. Two species of *Sargassum hemiphyllum* have been defined based on morphology and molecular data [1,9,10,11], namely, *S. hemiphyllum* var. *chinense* from China and *S. hemiphyllum* var. *hemiphyllum* from Korea and Japan.

*Sargassum hemiphyllum* is a dominant species in numerous subtropical marine communities, playing a crucial role in the construction of algal field habitats. It contributes to the reduction of eutrophication in aquatic environments, aids in the repair of damaged habitats, and supports the development of a comprehensive aquaculture model that encompasses multiple trophic levels. Macroseaweed beds of *S. hemiphyllum* are readily established because of their large size and rapid growth [12]. Additionally, *Sargassum hemiphyllum* possesses significant commercial value and can be utilized to extract medicinal and industrial raw materials, including seaweed polysaccharides, alginates, and marine biomass energy; this highlights its potential for economic development and utilization. However, their distribution and quantity in China are under threat owing to environmental pollution and climate change. Previous studies on *S. hemiphyllum* var. *chinense* have focused on artificial seedlings and transplantation, seedling growth, physiology, and biochemistry under different growth periods and stress conditions [13,14,15,16]. However, advances in genomics and transcriptomics have allowed for the study of nucleotide fragments from the nuclei, mitochondria, and plastids in taxonomic and phylogenetic research, thus outperforming traditional classification measures. For instance, Bi et al. [17] and Huang [18] demonstrated that the internal transcribed spacer (ITS) sequences of *S. hemiphyllum* var. *chinense* in the East and South Seas of China have similar morphological and genetic characteristics, while significant differences exist between *S*. *hemiphyllum* var. *chinense* in the Yellow Sea and those in other seas, indicating that it may be an independent geographical group. This finding supports the new classification system of *Sargassum*, which only retains the subgenera *Sargassum* and *Bactrophycus*. However, these studies are based on limited sequences and may not be capable of providing accurate information regarding phylogenetic relationships and taxonomy.

The complete organellar genome contains essential genetic information and additional phylogenetic signals, making it an effective molecular marker for distinguishing and identifying species and variants, thus allowing for more accurate phylogenetic analysis. Therefore, there is an urgent need for further comprehensive research on the topic. Owing to the small size and compactness of the organellar genome, coupled with the rapid development of high-throughput sequencing technologies [19,20], an increasing number of organellar genomes are published. However, compared with those for other *Sargassum* species, such as *S. polycystum* and *S. macrocarpum,* analysis of the omics data for *S. hemiphyllum* var. *chinense* remains limited [21,22,23,24,25].

Fucales, which includes *Sargassum,* is one of the three important orders of Phaeophyceae, along with Ectocarpales and Laminariales. The phylogenetic relationships among these orders remain controversial and poorly understood, with Fucales/Ectocarpales versus Laminariales or Fucales/Laminariales versus Ectocarpales established as sister groups. Peters and Burkhardt [26] proposed that the species in the order Fucales form independent branches, unlike other types of brown algae. Although Lee et al. [27] and Li et al. [28] constructed a phylogenetic tree by combining multiple genes to demonstrate the evolutionary relationships of the species belonging to the class Phaeophyceae, they did not provide a detailed discussion regarding the relationship between the orders Fucales, Ectocarpales, and Laminariales.

Studying and exploring the algal mitochondrial genome based on the endosymbiotic theory of eukaryotic evolution can provide a more in-depth understanding of the evolution of different eukaryotes. The endosymbiotic evolution of plastids in eukaryotic algae, including primary and secondary endosymbiotic events, has led to different evolutionary paths for eukaryotic algae groups. This suggests the existence of distinct “ancestors” and evolutionary events occurring at different times. Accordingly, research on organellar genomes can facilitate the elucidation of differences among eukaryotic hosts and provide insights regarding the evolution of different algal lineages. Furthermore, these data can inform the development of new molecular markers and screening of specific genes. 

In the study, we present the complete genome sequencing of the *S. hemiphyllum* var. *chinense* mitochondria and plastid. We compare the structure and evolutionary characteristics of the organelle genomes, focusing on genome structure, coding gene arrangement, structure–function relationships, and systematic evolution at the genus and order levels. We also present the codon preference, RNA editing sites, and repetitive sequences for the three important orders of Phaeophyceae: Fucales, Ectocarpales, and Laminariales. Collectively, our findings provide novel insights into the origin, evolution, and phylogeny of brown algae.

## 2. Results

### 2.1. Genomic Characteristics

The mtDNA and cpDNA of *S*. *hemiphyllum* var. *chinense* were assembled into single circular molecules of 34,686 and 124,323 bp, respectively (Figure 1). The GC contents were 36.55% (mtDNA) and 30.58% (cpDNA), with obvious AT shifts. This study was conducted using mtDNA and cpDNA separately (Table 1).

The mtDNA and cpDNA of *S. hemiphyllum* var. *chinense* encoded 65 genes (35 PCGs, 25 tRNAs, 3 rRNAs, and 2 ORFs) and 173 genes (137 PCGs, 28 tRNAs, 6 rRNAs, 2 ORFs, and 1 intron in *trn*L–UAA), respectively. Most tRNA genes have a typical cloverleaf secondary structure (Appendix A). There was no gene rearrangement or deletion except for that in *S. aquifolium*, which lacks the mitochondrial gene *trn*I (*uau*) and has a similar genome to the previously published organellar genomes of *Sargassum* species, indicating that the gene content and organization of *Sargassum* organellar genomes are highly conserved. The total length of intergenic spacers in mtDNA (1597 bp) was significantly shorter than that in cpDNA (19,859 bp), whereas the proportion of protein-coding regions was similar between mtDNA (75.98%) and cpDNA (75.34%). The content and total length of gene overlap regions in mtDNA (12,188 bp) were greater than those in cpDNA (675 bp), indicating a higher level of compactness in the former. The mtDNA and cpDNA of *Sargassum* species were relatively similar. Comparative genomics results showed that the size of the mtDNA of 14 *Sargassum* species ranged from 34,606–34,925 bp, with a GC content of 35.60–37.50%, while that of the cpDNA of five *Sargassum* species ranged from 124,075–124,592 bp, with a GC content of 30.40–30.60%. Intergenic spacers ranged from 1464–1835 bp and protein-coding regions accounted for 75.41–76.30% of the corresponding genome, 12 overlaps (*rps*8–*rpl*6, *rpl*6–*rps*2, *rps*2–*rps*4, *nad*1–*tat*C, *rps*12–*rps*7, *rpl*16–*rps*3, *rps*19–*rpl*2, *rps*13–*rps*11, *cox*2–*nad*4, *nad*5–*nad*6, *nad*3–*rps*14, and *rrn*5–*trn*M) of 1–90 bp between genes, and a total length of 81–211 bp. Thus, the primary difference was that there was no overlap between *nad*1–*tat*C of *S. yezoense* in the mtDNA of 14 *Sargassum* species. The total length of the gene intergenic spacer was between 17,349 and 21,938 bp, and protein-coding regions accounted for 75.00–75.45% of the corresponding cpDNA, five overlaps (*ftr*B–*ycf*12, *suf*C–*suf*B, *rps*1–thiS, *psb*C–*psb*D, and *rpl*23–*rpl*4) of 4–53 bp between genes, and a total of 75 bp. The only difference between *trn*A–*trn*I in *S. confusum* was in two 1 bp overlaps in the double copies. In summary, the cpDNA of Fucales species was much larger than mtDNA. The organellar genomes of *Sargassum* species were smaller with higher GC content, protein-coding regions occupied a larger ratio of the corresponding genome, and the total length of gene overlap was longer than that of non-*Sargassum* Fucales species mtDNA.

Six mtDNAs and seven cpDNAs were studied in Ectocarpales. *Pylaiella littoralis* of Acinetosporaceae has the largest mtDNA (58,507 bp), highest GC content (38.00%), longest intergenic sequence size (8030 bp), and most coding genes (36 PCGs and 16 ORFs), while *Colpomenia peregrina* of Scytosiphonaceae has the smallest mtDNA (36,025 bp), lowest GC content (32.00%), and shortest intergenic sequence size (1499 bp). Other species had highly similar genome sizes of 36,918–38,419 bp, GC content of 32.90–34.40%, and an intergenic sequence size of 2221–4134 bp. Among the six cpDNAs of Ectocarpales, *Ectocarpus siliculosus* had the largest cpDNA (139,954 bp), longest intergenic sequence length (28,995 bp), highest number of coding genes (185 genes, including 142 PCGs, 30 tRNAs, 6 rRNAs, and 6 ORFs), and the least protein-coding sequences (69.20%). The size of other species cpDNA was between 133,508 and 138,815 bp, GC content was 29.80–31.30%, encoded proteins accounted for 69.20–72.40%, and the total gene interval was 23,197–26,578 bp, with consistent overlapping fragments (65 bp) located in *rpl*23–*rpl*4 (8 bp), *psb*D–*psb*C (53 bp), and *ycf*16–*ycf*24 (4 bp). Gene numbers varied owing to the tRNA and ORFs. Mitochondrial gene numbers ranged from 64–69 with 35 PCGs, 3 rRNAs, 24–25 tRNAs, 2–7 ORFs, and 177–180 plastid genes, of which 140 PCGs (*Pleurocladia lacustris* lacks *rpl*32), 2–5 ORFs, 3 rRNAs, 27–30 tRNAs, and a special transfer–messenger RNA (tmRNA) with both tRNA and mRNA properties existed in individual species of Ectocarpales (*Scytosiphon canaliculatus*, *Scytosiphon promiscuus*, *Endarachne binghamiae*, and *Cladosiphon okamuranus*).

Among the 12 mtDNAs and 8 cpDNAs in Laminariales, *Lessonia spicata* (Lessoniaceae) had the smallest mtDNA (37,097 bp), smallest intergenic sequence length (1861 bp), and longest protein-coding sequence length (73.21%), while *Costaria costata* (Agaraceae) had the smallest cpDNA (129,947 bp). Other organellar genomes were relatively similar, with the mtDNA size ranging from 37,366–38,007 bp, GC content 32.00–35.40%, intergenic sequence 2107–2969 bp, and total number of genes 63–67, primarily composed of 23–25 tRNAs, 2–3 rRNAs, and 2–3 ORFs. The cpDNA size was 130,305–130,784 bp, with a GC content of 30.60–31.10%. The protein-coding sequence accounted for 72.67–73.52%. Gene numbers (171–173) varied with tRNAs (25–27). The total length of gene intervals was between 21,426 and 22,249 bp, and that of the overlapping sequences was 71 bp. The latter was located in four regions, *ycf*12–*ftr*B (6 bp), *rpl*4–*rpl*23 (8 bp), *ycf*24–*ycf*16 (4 bp), and *psb*D–*psb*C (53 bp and 8 bp in *Laminaria digitata*), excluding the 26 bp unique to *Laminaria digitata* (Laminariaceae).

### 2.2. Organellar Gene Analysis

The arrangement of the mtDNA coding genes of *S. hemiphyllum* var. *chinense* is consistent with that of the other 13 studied *Sargassum* species. Fucales, Ectocarpales, and Laminariales share a consistent gene configuration; however, each order has distinct characteristics, except for occasional gene loss in certain species. The coding genes between mitochondrial genes *nad*4 and *nad*5 differed among the three orders, namely, *trn*I or *trn*K of Fucales, *trn*S or *trn*Q of Ectocarpales, and *trn*K or *trn*X of Laminariales, whereas the gene between mitochondrial genes *trn*W and *trn*Q of the species of the three orders was *trn*M or *trn*I. Notably, the collinearity among the 37 species of the three orders was highly consistent, but the gene configuration can be explained by the arrangement of the two tRNA genes, *trn*A and *trn*D (Figure 2 and Appendix A).

The arrangement of the cpDNA coding genes of *S. hemiphyllum* var. *chinense* is consistent with that of the other four *Sargassum* species that have been published; it is the same as that in other Fucales species, except for the deletion of tRNA. Among the cpDNA coding genes of six species studied in Ectocarpales, in addition to *Scythosiphon canaliculatus*, *Scythosiphon promiscuous*, *Endarachne binghamiae*, and *Cladosiphon okamuranus*, the same configuration of coding genes encodes one more tmRNA; the 10.8 kb coding gene ranges between *psb*Y and *rbc*S of *Ectocarpus siliculosus* (encodes one more *psb*A and *rpl*32) and *Pleurocladia lacustris* (gene arrangement is reversed compared to that of other species) and is scattered. Eight cpDNAs of Laminariales have been published with the same configuration of coding genes except for the missing genes of individual species. The arrangement of organellar coding genes in Fucales, Ectocarpales, and Laminariales clearly distinguishes the three orders. The first half of the cpDNA displayed a consistent arrangement between Fucales and Laminariales, whereas the second half was similar to that of Ectocarpales. The collinearity difference between orders was higher than that within orders, indicating that cpDNAs of different order species have different evolutionary patterns; numerous gene deletion (horizontal gene transfer) and gene rearrangement events occurred, showing their unique evolutionary patterns (Figure 2 and Appendix A). 

### 2.3. PCGs

The mtDNA of 14 *Sargassum* species encoded 35 PCGs with the standard genetic code and the same start codon, ATG, of which five genes (*rpl*2, *rpl*16, *rps*3, *rps*19, and *tat*C) were encoded by the light chain. Genes with the same length and termination codon accounted for 37.14% of all genes. The cpDNA of the five *Sargassum* species all encoded 137 PCGs, of which 57 genes were encoded by the light chain and 136 genes used ATG as the start codon, except *psbF*, which used GTG; the proportion of genes with the same length and termination codon was 80.29% (Appendix A).

We analyzed the composition and structural characteristics of the organellar genomes to further explore the evolutionary relationships among the three representative orders. All mtDNA encoded 35 PCGs, and the mtDNA of *Pylaiella littorali* in Ectocarpales encoded more than one *rpo* gene. However, there were significant differences in cpDNA. Fucales, Ectocarpales, and Laminariales encoded 137, 140, and 138 PCGs, respectively, whereas the latter two orders obtained *syfB* and *ycf*17. Notably, more than one *rpl*21 was encoded by the cpDNA of Ectocarpales, and the cpDNA of Laminariales lacked *pet*L (Figure 3).

### 2.4. PCG Codon Use

In *S*. *hemiphyllum* var. *chinense*, 35 mtPCGs and 137 cpPCGs encoded 8780 and 31,111 amino acids, respectively. Leu (13.05%/mtDNA and 10.78%/cpDNA) and Ile (7.77%/mtDNA and 9.83%/cpDNA) were the most frequently used amino acids, whereas Cys (1.6%/mtDNA and 0.89%/cpDNA) was uncommon in both the mitochondria and plastids. In addition, Ser (7.75%), Val (7.07%), and Gly (7.04%) were the most frequently encoded amino acids in mtDNA, compared with those in cpDNA, which only Lys (8.09%). Trp (1.11%), His (1.61%), and Met (1.96%) were most frequently identified in cpDNA, whereas Trp (1.66%) was most frequently identified in mtDNA (Figure 4). The RSCU value of the *S. hemiphyllum* var. *chinense* organellar genome is another aspect that we investigated by analyzing the codons of PCGs (excluding termination codons). Five thousand six hundred and thirty-nine codons (63.99%) in mtDNA and 23,986 codons (79.10%) in cpDNA had RSCU values > 1.0, indicating that they were used more frequently than synonymous codons.

We analyzed the codon preference of the organellar genomes of Fucales, Ectocarpales, and Laminariales, represented by *S. hemiphyllum* var. *chinense*, *Ectocarpus siliculosus*, and *Saccharina japonica*, respectively.

The GC content of the third codon of the base was significantly lower than that of the first and second codons, indicating that the third codon was primarily composed of A/U and the highest ENC value of Fucales was 47.12 (mtDNA) and 39.44 (cpDNA), indicating that the organellar genomes of Fucales have weak codon preferences and relatively low expression levels compared with those of the other two orders (Table 2). The neutrality plot and ENC-plot analyses showed that natural selection was the main influencing factor of organellar gene codon use preference in the three orders. Notably, *rpl*6 and *nad*4 in the mtDNA and *ycf*33 and *ycf*65 in the cpDNA of Fucales were mainly affected by mutations (Appendix A). The PR2-plot analysis showed that the distribution of mtPCGs in the four planes was not uniform, and most of the mtPCGs in the three orders were distributed in the lower region, mainly in the lower right. Therefore, the usage frequency of T and G at the third base of the gene codon was higher than that of A and C, especially in Fucales (27/35). However, the distribution of cpPCGs in the three orders was consistent, and most of them were located at or near the 0.5° horizontal line, indicating that base mutation plays an important role in the codon preference of plastids in the three orders (Appendix A).

The analysis of RSCU showed that the proportion of high-frequency codons for three orders (RSCU > 1.00) ranged from 37.7 to 44.26%, and the proportion of the third base A/U in high-frequency codons was 95.65–100% in addition to the high-frequency codons in mtDNA of Fucales, which have partial G/C endings, consistent with the codon preference of most species ending with A and U. CUU was a common high-frequency codon in the mtDNA of the three orders, UUG was unique to Fucales, GCA was unique to Ectocarpales, and GGG was unique to Laminariales. No preferences were noted for AUG and UGG. Furthermore, there was a clear bias toward purines at the third codon position for all codons. The codons AGC and UUA had the lowest and highest RSCU values, respectively. Codons AUG and UGG showed no bias (RSCU = 1). Meanwhile, the numbers of codons in the higher (RSCU > 1) and lower (RSCU < 1) regions were relatively similar across different orders (Appendix A). The optimal number of mtDNA codons was 11 (Fucales), 13 (Ectocarpales), and 14 (Laminariales), with the consensus UUA and GUU and the unique UUG, CCC, UAA, and CAU codons in Fucales. For cpDNA, the optimal codons were 11 (Fucales), 20 (Ectocarpales), and 14 (Laminariales), with the consensus of UUA, GUU, UCU, GAA, and GGU; only CCA was found in Fucales (Figure 5).

### 2.5. Identification of Potential RNA Editing Sites in PCGs

RNA editing, specifically insertion, deletion, or replacement of nucleotides in the coding region of messenger RNA, is widespread in plants other than mosses. We predicted 80 and 20 RNA editing sites in 35 mtPCGs and 137 cpPCGs, respectively, in the organellar genome of *S. hemiphyllum* var. *chinense* (Table 3). The largest numbers of editing sites were on *nad*2 and *nad*8, which accounted for 10% of the total editing sites of mtPCGs, and *nad*4 was the secondary editing site, accounting for 8.75% (Figure 6). The first position encoded by these three genes (triplet code) contained 32 RNA editing sites, accounting for 40.0%, whereas the second position accounted for 60.00%. The hydrophilicity of 62.5% of the amino acids did not change; 16.25% of amino acids became hydrophilic, and 21.25% became hydrophobic. Of the 80 codons edited in the mitogenome, 27.5% were proline codons: 13 were changed into serine, and 9 were changed into leucine. The second most important editing event (27.5%) occurred in alanine codons, which were all changed into valine codons. Similarly, out of the 20 edited codons in the plastome, threonine codons and alanine codons were the most frequently edited (7/20, 6/20), leading to the conversion of this amino acid to isoleucine and valine. The largest number of RNA editing sites on *rps*4 (5) accounted for 26.32%, and *rps*7 was the secondary editing site (Figure 6). The first position of the triplet code contained two sites, accounting for 10.53%, whereas the second position contained 89.47%. The hydrophilic properties of 36.84% of the amino acids in the cpDNA did not change; 5.00% of amino acids became hydrophilic, and 50.00% became hydrophobic. The phenomenon of 27.5% of codons in mtDNA becoming valine and 36.84% of codons in cpDNA becoming Ile indicates that Val and Ile were the main amino acids in the predicted editing codon.

We predicted the RNA editing sites of shared mtPCGs and cpPCGs in Fucales, Ectocarpales, and Laminariales to identify similarities. Each species had its own unique RNA editing sites in comparison with other species, which indicates that RNA editing sites are independently lost after species divergence (Appendix A). The selection of RNA editing sites exhibited a high compositional bias, and all RNA editing sites were of the C-to-T editing type. The RNA editing sites in cpDNA were all located in genes encoding ribosomal proteins (*rps* and *rpl*), with the most common being on *rps*4; no RNA editing sites were found on *rps*14 in the Fucales species. In comparison, mtPCGs had more RNA editing sites than cpPCGs, and the transcripts of NADH dehydrogenase subunits and cytochrome c biogenic genes were significantly edited, in addition to ribosomal proteins, among which *nad*4L was only edited in Ectocarpales species. The RNA editing types of mtPCGs and cpPCGs were essentially the same: 28 codon transfer types corresponded to 12 amino acid transfer types, CTT(L)→TTT(F) and GCT(A)→GTT(V) were common in mtPCGs, and ACA(T)→ATA(I) and ACT(T)→ATT(I) were common in cpPCGs. There were 28 types of RNA editing present in the plastome. Compared with the order Ectocarpales and Laminaria, the order Fucales did not have CGC (R)=>TGC (C), CTC (L)=>TTC (F), CCC (P)=>CTC (L), ACG (T)=>ATG (M), TCG (S)=>TTG (L), TCT (S)=>TTC (F), ACC (T)=>ATC (I), and CCG (P)=>TCG (S).

### 2.6. Repeat Sequence Analysis

A preliminary survey of the genome of the *S. hemiphyllum* var. *chinense* identified simple sequence repeat (SSR) domains as the most abundant repeat types in the plastome. SSRs (or microsatellites) are 1–6 bp sequence repeat units. SSRs are particularly useful owing to their polymorphisms, easy PCR detection, co-dominant inheritance, and extensive genome coverage. We identified 35 SSRs (12/mtDNA and 23/cpDNA) in *S. hemiphyllum* var. *chinense*. Single-nucleotide repeat sequences are the most common SSR type and account for 100% (mtDNA) and 65.22% (cpDNA). A single trimeric SSR was located in the plastid gene *rrn*5–*ycf*19 (Appendix A). Twenty interspersed repeat sequences with a length of 30 bp or longer were detected in the organellar genome of *S. hemiphyllum* var. *chinense*, of which only one 33 bp forward repeat was located on the mtDNA, and the remaining 19 were on the cpDNA. Among the 19 non-serial repeats, 17 were palindromic repeats, 1 was a direct repeat, and 1 was a reverse repeat; the longest palindrome repeat sequence was 5446 bp in size (Appendix A). Tandem repeats are core repeat units of 1–200 bp that are repeated several times in series. Only nine tandem repeats of 14–24 bp length were observed in the cpDNA of *S. hemiphyllum* var. *chinense* (Appendix A).

We used *Ectocarpus siliculosus* and *Saccharina japonica* as representatives to analyze the repeat sequences of the organellar genomes of Ectocarpales and Laminariales. The mtDNA of Ectocarpales had 12 SSRs (including mononucleotide and dinucleotide repeats), two (30 and 62 bp) scattered duplications, and one tandem duplication. For cpDNA, there were 18 SSRs with one 4-mer nucleotide repeat and 24 interspersed repeats, totaling 19,670 bp, with the longest palindrome of 8616 bp and eight tandem repeats. The SSRs of mtDNA in Laminariales contained only seven mononucleotide repeats, two (32 and 40 bp) interspersed repeats, and no tandem repeats. There were 16 SSRs in the cpDNA with four dinucleotide repeats and one trimeric repeat, 43 interspersed repeats totaling 22,136 bp with four palindromic sequences larger than 1000 bp, and eight tandem duplications.

### 2.7. SNP Analysis

According to the currently recognized subgenera, namely *Sargassum* and *Bactrophycus*, 197 SNP sites (89 non-synonymous and 108 synonymous mutations) belonging to the two subgenera were found in 35 PCGs and two ORFs of mtDNA in 14 *Sargassum* species, except *atp*6, *nad*11, and *rps*14 (Figure 7A).

According to the Sect. level (Sect. is an abbreviation for Section) identification based on the current morphological and molecular data, *S. hemiphyllum* var. *chinense*, *S. confusum*, *S. mucium*, and *S. thunbergii* belong to Sect. *Teretia*; Sect. *Sargassum* includes *S. vacellianum*, *S. fluitans*, *S. natans*, and *S. spinuligerum*, and other subgroups have only one species each. Of the 197 SNP sites, 115 (49 non-synonymous and 66 synonymous mutations) were identified as Sect. *Teretia*; 74 (21 non-synonymous and 53 synonymous mutations) were identified as Sect. *Sargassum*; and no sites were identified on *atp*9, *rps*8, *rps*2, *nad*4L, *rpl*5, *rps*3, *rps*19, *atp*6, *nad*9, *rps*14, and *atp*8 (Figure 7B,C).

At the species level, 35 PCGs and two ORFs of 14 *Sargassum* species and 3040 SNP sites (Figure 7D) that could be used for single-species identification were compared, of which 360 SNP sites (216 synonymous mutations) could be used to identify *S. hemiphyllum* var. *chinense*. In *Sargassum* species, most SNP sites were distributed on the *cox*1 gene, followed by *nad*5 and *nad*4 (Figure 7E). In summary, two genes (*rps*7 and *cox*2) were used as SNP sites for identifying the subgenera, Sects., and species of *Sargassum* (Figure 7F,G).

### 2.8. Phylogenetic Analysis and Ka/Ks

To determine the evolutionary status of *Sargassum*, a phylogenetic analysis was performed based on 35 PCGs from 27 published mtDNAs and 137 PCGs from 18 published cpDNAs of *Sargassum* with *Coccophora langsdorfii* as the outgroup (Figure 8). The topological structures of mtPCGs and cpPCGs were consistent, and the posterior probabilities for the other branches were 1.0. *Sargassum* species can be clearly divided into two branches: subgenus *Bactrophycus*, which includes *S. hemiphyllum* var. *chinense* and subgenus *Sargassum*. *Bactrophycus* is clustered into two main branches, and *S. hemiphyllum* var. *chinense* aggregates with *S. kjellmannianum*, *S. muticum*, and *S. conflosum*, forming Sect. *Teretia*, followed by *S. fusiforme* (Sect. *Hizikia*) and *S. horneri* (Sect. *Spongocarpus*), and then aggregates with Sect. *Halochloa* to form sister branches. Sects. *Polycystae* and *Ilicifolia* of *Sargassum* converged into one branch and Sects*. Sargassum* and *Zygocarpicae* converged with them; finally, Sect. *Binderiana* converged on the outermost branch. This supports the classification of *S. phylocystum, S. mcclurei*, and *S. henslowianum* as *Ilicifolia*, while *S. ilicifolium* var. *concomitatum* does not support Sect. *Ilicifolia* and appears to have a closer genetic relationship with *Zygocarpicae*.

Phylogenetic analysis based on published mtDNAs and cpDNAs was performed to further explore the evolutionary relationship of the three representative orders with the outgroup *Dictyopteris divaricata*. Classification of the cluster tree based on the RSCU values showed that Fucales evolved independently of Ectocarpales and Laminariales, thus self-declaring a branch, whereas the clades of the latter evolved intermittently within each other (Appendix A). The complete nucleotide sequences of the organellar genomes did not distinguish the outgroup, and the differences in the topology of the phylogenetic trees constructed from mtDNAs and cpDNAs were insufficient to fully explore the evolutionary history (Appendix A). Results of the shared mtPCGs showed that Ectocarpales and Laminariales first gathered into one branch, then gathered into a larger branch with Fucales, and finally gathered with the outgroup that included *Dictyopteris divaricata*. The phylogenetic tree of cpPCGs was consistent with the topological structure derived from the mtPCGs. The overall structures of the RSCU cluster plots based on mtDNA and the phylogenetic tree constructed using shared PCGs were consistent.

To evaluate the selection pressure of PCGs in the evolutionary dynamics among closely related species, the substitution rates (Ka/Ks) of 35 mtPCGs from 36 species and 133 cpPCGs from 21 species of Fucales, Ectocarpales, and Laminariales were compared using *S. hemiphyllum* var. *chinense* as a reference (Figure 9). The Ka/Ks of the mtPCGs ranged from 0.014 at *atp*9 in *Scythosiphon romanaria* to 1.216 at *rpl*31 in *Cladosiphon okamuranus*, 0.004 at *psb*C in *Coccophora langsdorfii*, and 3.604 at *rpl*2 in *Fucus vesicosus*. The Ka/Ks value of the specific replacement rate of *S. hemiphyllum* var. *chinense* organellar genes was less than 1, which indicated that most genes experienced negative selection during the evolutionary process (except in *Colpomenia peregrina*, *Endarachne binghamiae*, *Cladosiphon okamuranus*, *S. thunbergii*, *Coccophora langsdorfii*, and *Fucus vesculosus*). However, the Ka/Ks value of *cox*2 between *S. hemiphyllum* var. *chinense* and *Colpomenia peregrina* was 1.029 and between 1.170 and 1.215 for *rpl*31 for *S. hemiphyllum* var. *chinense*, *Endarachne binghamiae*, and *Cladosiphon okamuranus* in mtPCGs. There are two substitution genes (*rpl*12, 2.864; *rpl*24, 1.165) with higher Ka/Ks values between *S. hemiphyllum* var. *chinense* and *S. thunbergii*, one substitution gene, *rpl*2 (3.604), between *S. hemiphyllum* var. *chinense* and *Fucus vesicosus*, and one substitution gene, *ycf*33 (1.673), between *S. hemiphyllum* var. *chinense* and *Coccophora langsdorfii* in cpDNA. These findings indicate that positive selection existed. In addition, *atp*9 and *cox*1 had the lowest average Ka/Ks values among mtPCGs (0.034 and 0.050, respectively), while *pet*N and *pet*B had the lowest average Ka/Ks values among cpPCGs (0.004 and 0.007, respectively), indicating strong purification selection. Overall, organellar genes are highly conserved during the evolution of Phaeophyceae species.

## 3. Discussion

### 3.1. Genomic Characteristics

Comparative analysis of the gene structures of organelles in *Sargassum* showed that the genome structure, including gene number, gene arrangement, overlapping regions between genes, and the total length of gene interval regions, was highly similar, which is consistent with previously reported results in *Sargassum* species [29,30]. The near collinearity of *Sargassum* organellar genomes is hardly surprising, considering the organellar genomes of Fucales, Ectocarpales, and Laminariales have maintained a conserved gene order and content despite millions of years of separate evolutionary histories. In contrast to the previously registered mtDNA of *S. hemiphyllum* (NC024861), a total of seven identified SNP sites of *S. hemiphyllum* var. *chinense* from different geographical sources were found, with more SNP sites remaining to be further analyzed.

Analysis of organellar genomes among Fucales, Ectocarpales, and Laminariales species revealed 25 core mtPCG sets and 133 core cpPCG sets, with most PCGs using ATG as the start codon and only the plastid gene *psb*F using GTG as the start codon. The start codon GTG is mainly used in specific plastid genes, such as *psb*F, which indicates the evolutionary conservation of these genes and associated processes [31]. Differences in gene expression levels are mainly related to tRNA, an essential component of protein translation, and play a decisive role in cell growth, proliferation, and organ development [32]. Gene loss or acquisition is a common event in the evolution of cpDNA and is closely related to natural classification. *ycf*17 and *syf*B are present in Ectocarpales and Laminariales, with *pet*L only absent in Laminariales species. The *ycf17* gene encodes a small protein of the early light-inducible protein family with a single membrane-spanning helix, which has been suggested to belong to the extended LHC antenna protein superfamily [33,34]. *syf*B encodes phenylalanine-tRNA ligase *β*, whose subunit structure and function have not been reported, which complicates further characterization. Although genes can be transferred to the nucleus through endosymbiotic gene transfer [35], we could not find a homologous sequence of the deleted gene in the corresponding target nuclear genome because no corresponding nuclear genome was available. Therefore, we inferred that the deleted gene in the cpDNA may be homologous to the nuclear gene. Analysis of gene arrangement and collinearity showed that the mtDNA of the three orders was highly similar; homologous genes were arranged in a consistent manner and had a relatively high collinearity relationship, showing a relatively conservative evolutionary model. In the mtDNA of the order Fucales and Ectocarpales species, the genes *trn*D and *trn*A are closely arranged, but their positions of rearrangement differ; the arrangement is *trn*S-*trn*D-*trn*A-*rps*10 in Fucales species, while in Ectocarpales, it is *trn*K-*trn*D-*trn*A-*trn*V. In contrast, *trn*D and *trn*A are dispersed throughout the mtDNAs of the order Laminariales species, with *trn*A positioned between *trn*K and *trn*V and *trn*D located between *trn*S and *rps*10. Overall, the *trn*D–*trn*A identified in this study represents a rearrangement event of mitochondrial genes within the class Phaeophyceae. This finding is both unique and representative among the three different orders of this class, thereby enhancing our understanding of mitochondrial gene rearrangement in Phaeophyceae. cpDNA showed only a relatively conservative evolutionary model at the order level, and the collinear relationship between orders was poor, suggesting a unique evolutionary model for each order. Generally, gene rearrangement is caused by the insertion of repeat sequences, and abnormal recombination is common [36]. It is speculated that the latter plays a leading role in the gene arrangement of the three orders when combined with the analysis of repeat sequences.

### 3.2. Codon Use Preferences

Phylogenetic and evolutionary analyses based on codon use preferences have not been extensively explored in algae, it provides valuable means for evolution by selection and mutation at the molecular level [25,27,37,38,39]. In the organellar genome of the genus *Sargassum*, the codons UUA and GUU, which correspond to leucine and valine, respectively, exhibited higher relative synonymous codon usage (RSCU) values exceeding 2.0. Additionally, in the chloroplast DNA (cpDNA), the RSCU value for AGA, which codes for arginine, also surpassed 2.0. These synonymous codons were utilized more frequently than anticipated, indicating a more stable usage pattern in the chloroplast genome of *Sargassum* compared to the mitochondrial genome; this is consistent with previous reports on the use of synonymous codons in some of the organelle genomes of *Sargassum* [27]. Notably, when compared to other orders of Phaeophyceae, distinct changes in synonymous codon usage patterns were observed in both the chloroplast and mitochondrial genomes. These specific patterns may serve as valuable indicators for studying the evolutionary dynamics of the brown algae family in future research. Comparative analysis of codon usage shows that the regression slope of the mtPCGs of *S. hemiphyllum* var. *chinense* is 0.0264, indicating that the effect of mutation pressure is only 2.64%, whereas the negative correlation between GC12 and GC3 of cpPCGs shows that the mutation pressure accounts for 13.10%, indicating that mutation pressure plays a secondary role in codon use bias and that plastids are more affected by mutation than mitochondria. The data points of most organellar PCGs of *S. hemiphyllum* var. *chinense* were located below the ENC-plot standard curve, indicating that the codon usage bias of these PCGs was mainly affected by natural selection in the process of species evolution [40], which is consistent with the results of the neutral analysis. Most of the high-frequency codons in mtPCGs (16/18) and cpPCGs (24/24) of *S. hemiphyllum* var. *chinense* end with A/U, indicating that nucleotide composition plays an important role in codon use. To evaluate the relationship between the level of gene expression and codon preference, we further analyzed the best codon of *S. hemiphyllum* var. *chinense*; 11 codons were defined as the best codon in cpPCGs ending with A/U, which is different from mtPCGs. The results for Ectocarpales and Laminariales were similar to those for Fucales. The third position of the codon has the same A/U bias, but the high-frequency codon and the optimal codon show significant differences, indicating that the optimal codon has a specificity and preference for different species. The optimal codons of mtPCGs and cpPCGs of the same species have large differences, indicating that mtPCGs and cpPCGs preferences in the same species are inconsistent, which may be related to the evolutionary process of different organelles.

### 3.3. RNA Editing Sites

RNA editing occurs after transcription and is conducive to protein folding. The number of RNA editing sites varies among species. This converts a sense codon into a more evolutionary conserved one. Comparative analysis of RNA editing sites showed that 80 RNA editing sites were found in 35 mtPCGs, and only 20 RNA editing sites were found in 137 cpPCGs, which were significantly less than those in higher plants (*Arabidopsis* contains 441 RNA editing sites in 36 genes, and rice contains 491 RNA editing sites in 34 genes) [41]. However, the high frequency of RNA editing in Fucales, which was nearly 14.3% greater than in the mitogenome of Ectocarpales and Laminariales, indicates that the frequency of RNA editing has fluctuated during Fucales diversification. The total number of plastid RNA editing sites is approximately one-quarter (0.25) that of the mitochondria, and the RNA editing sites in the plastid only occur in the synthesis of ribosomal proteins. A striking discrepancy in RNA editing between the two organellar genomes is found in *S. hemiphyllum* var. *chinense*, in which 80 mitochondrial editing sites coexist with 20 such sites in chloroplast. All RNA editing sites were of the C-to-T editing type, consistent with the fact that C-to-T is the most common editing type in plant organellar genomes [42]. Identification of the RNA editing site provides key information for predicting the function of a gene containing a new codon. Whether the number of RNA editing sites positively correlates with the size of the organellar genome requires further investigation. Overall, the codon preference of targets for RNA editing and site distribution showed similar trends across species.

### 3.4. Repeat Sequence

Non-tandem and tandem repeat SSRs were the primary targets of this study. Repetitive sequences have been shown as crucial for intermolecular recombination in previous mitochondrial studies. Therefore, repetitive sequences play a pivotal role in the assembly of organellar genomes [43]. Comparative analysis of repeat regions showed more repetitive sequences in the cpDNA of *S. hemiphyllum* var. *chinense* than in mtDNA, which may be evidence of frequent intermolecular recombination, which dynamically changed the structure and conformation of the organellar genome during evolution. A comparison of the cpDNA and the mtDNA assembled from the same dataset revealed nine mitochondrial DNA fragments with sequences similar to those in the repeat regions of the cpDNA, suggesting that the repeat regions might be transferred into the mitogenome. This study provides valuable information on the taxonomic classification and molecular evolution of members of the genus *Sargassum*. We also compared the organellar genomes of *S. hemiphyllum* var. *chinense* with those of other brown algae to better understand their structure and organization. In summary, the organellar genomes of *S. hemiphyllum* var. *chinense* have characteristics that are common to those of other brown algae. In the mtDNA and cpDNA of Fucales, repeat sequences account for 0.6% and 10.78%, respectively, which are considered low proportions of repeat [44,45] and lower than those of *Ectocarpus siliculosus* and *Saccharina japonica*. The difference in the proportion of repeat sequences in the organellar genomes may be because the organelles of *S. hemiphyllum* var. *chinense* contain mainly short repeat units, while Ectocarpales and Laminariales contain mainly long repeat units, indicating that the expression level and evolutionary pressure of Fucales species are relatively low, and their environmental adaptability is stronger. Phylogeographic and repetitive sequence analysis based on the complete cpDNA and mtDNA indicated that the novel markers, particularly cpSSRs, could provide a more detailed picture of *Sargassum* population structure in order Fucales, which requires further in-depth research.

### 3.5. Phylogenetic Analysis and Ka/Ks

The substitution rates (Ka/Ks) of 35 mtPCGs and 133 cpPCGs in 21 species of the order Fucales, Ectocarpales, and Laminariales were almost no non-synonymous changes, indicating that in most cases, the genes of the class Phaeophyceae species were purified and selected to eliminate harmful mutations while keeps the protein unchanged. These results will help lay the foundation to better understand the evolutionary relationships among Fucales, Ectocarpales, and Laminariales. The latest classification of *Sargassum* is based on a combination of morphology and molecular biology; subgenera with no obvious differentiation of stems and leaves were categorized as *Phyllorichia*, whereas those with obvious differentiation of stems and leaves with retroflex were categorized as *Bactrophycus*, and those with non-retroflex were categorized as *Sargassum*. This study supports the retention of the subgenera *Bactrophycus* and *Sargassum* in the genus *Sargassum* and *S. hemiphyllum* var. *chinense* in Sect. *Teretia* in subgenus *Bactrophycus*, which is consistent with previous studies [11,18,46,47]. Based on existing organellar genomes, the genus *Sargassum* was divided into two groups, namely Sect. *Halochloa* and Sect. *Teretia*, which are combined by Sects. *Hizikia*, *Teretia*, and *Spongocarpus*. The subgenus *Sargassum* is divided into three groups: Sects. *Polycystae* and *Ilicifolia* merged into the first group, Sects. *Sargassum* and *Zygocarpicae* merged into the second group and Sect. *Binderiana* formed its own branch. *Sargassum phylocystum, S. mcclurei,* and *S. henslowianum* merged with *S. Ilicifolium*, indicating that they are closely related. It is recommended to move the other three species from Sect. *Phyllocystis* of subgenus *Bactrophycus* to Sect. *Ilicifolia* of subgenus *Sargassum*, while *S. ilicifolium* var. *concomitatum* should be clustered with *S. graminifolia* and classified as Sect. *Zygocarpicae*. There is one thing worth paying attention to *Sargassum siliquastrum*, *Sargassum macrocarpum*, and *Sargassum serratifolium* are observed on the same branch and node with high support rates in Figure 8A. However, in the phylogenetic trees depicted in Figure 8B,C, these species are located on different nodes of the same branch with high support rates. It is speculated that analytical factors (such as an insufficient number of genes or loci) and biological factors (such as incomplete lineage sorting) play a certain role in the evolution of *Sargassum siliquastrum*, *Sargassum macrocarpum*, and *Sargassum serratifolium*. This result provides a summary of the actual classification and the identification key for the subdivision of *Sargassum*.

The conservation within the *Sargassum* genus is supported by phylogenetic analysis, which revealed that all members of *Sargassum* are grouped into a single clade (Fucales) according to their taxonomic order. This consistent clustering suggests that there has been no rapid evolution of the *Sargassum* genus concerning organellar genomes. Notably, *S. kjellmannianum* and *S. muticum* exhibited extremely high sequence similarity and were always identified as sister species phylogenetically in both chloroplast and mitochondrial genomes. However, the number of *Sargassum* species reaches up to 360, but the proportion of organellar genomes published is less than 10%, which severely limits the identification and phylogenetic analysis of *Sargassum* species. Therefore, with the rapid development of high-throughput sequencing technology and molecular systematics, the acquisition of more organelle genome data for this genus will help to make more objective judgments on species identification and its phylogenetic evolution.

We reconstructed a phylogenetic tree based on the 165 PCGs (132 chloroplast PCGs and 32 mitochondrial PCGs) shared in 50 species of the three orders and further verified their phylogenetic relationships, but we did not consider their morphological characteristics or other genetic factors. The overall topological structure established using plastid and mitochondrial genomes was consistent. The species in the order Fucales, Laminariales, and Ectocarpales form independent branches, and the first two form a large branch; this means that the differentiation routes of Laminariales and Ectocarpales in the molecular evolutionary tree were more similar than those of Fucales, and they appeared as nodes on the same branch, indicating that Laminariales and Ectocarpales are more closely related, forming an evolutionary branch together, whereas Fucales evolved relatively independently.

## 4. Materials and Methods

### 4.1. Sampling and DNA Extraction

*S. hemiphyllum* var. *chinense* was collected from the Zhongpeng Island intertidal zone, Shantou City, Guangdong Province, China, on 13 May 2017 (117°17′52″ E, 23°16′41″ N). Subsequently, the algae were transported to the laboratory in an insulated box, maintaining a temperature range of 20 °C to 26 °C. They were then cultured in sterile filtered seawater for 3 d under a fluorescent light (80–110 μmol photons m^–2^ s^–1^; 12 h light/dark cycles) at 24–26 °C in the laboratory. High-quality DNA was extracted from 5 g of fresh tissue using a modified cetyltrimethylammonium bromide (CTAB) method [48] after thoroughly cleaning the algae with sterile seawater using a sterile brush and being dried with paper towels. DNA concentration, purity (260/280 nm and 260/230 nm), and integrity were measured using a Qubit 3.0 fluorometer (Thermo Fisher, Waltham, MA, USA) and a NanoDrop ND1000 ultraviolet spectrophotometer (Thermo Fisher, Waltham, MA, USA) on 1% agarose gel with λDNA/HindⅢ as the marker. The remaining clean materials and unused DNA (sampler: 2017050129) were stored at −20 °C in the Joint Laboratory of Large Seaweed at Xianmen University.

### 4.2. Organellar DNA Sequencing and Genome Assembly

After DNA isolation, 1 μg of purified DNA was fragmented and used to construct short-insert libraries (insert size: 430 bp) according to the manufacturer’s instructions and sequenced using the Illumina Hiseq 4000 (Illumina, Inc., San Diego, CA, USA) [49]. Prior to assembly, raw reads with adaptors were filtered for a quality score below 20 (Q < 20), for those that contained a proportion of uncalled bases (“N” characters) ≥ 10%, and for duplicated sequences.

mtDNA/cpDNA was reconstructed using a combination of *de novo* and reference-guided assemblies and assembled [50]. First, the filtered reads were assembled into contigs using SOAPdenovo (version 2.04) [51]; the best assembly result was obtained after splicing multiple k-mer parameters using ABySS software (version 2.0.2; http://www.bcgsc.ca/platform/bioinfo/software/abyss) (accessed on 3 July 2020). Second, the contigs were aligned to the reference genome of NC024861/MT800998 using BLAST, and the aligned contigs (≥80% similarity and query coverage) were ordered. Third, clean reads were mapped to the assembled draft of mtDNA/cpDNA to correct the wrong bases and fill the gaps through local assembly using GapCloser (version 1.12; https://sourceforge.net/projects/soapdenovo2/files/GapCloser/) (accessed on 3 July 2020).

### 4.3. Genome Annotation

Mitochondrial genes were annotated using homology alignments and de novo prediction, and the gene set was integrated using EVidenceModeler software (version 1.1.1) [52]. tRNA and rRNA genes were predicted using tRNAscan-S and rRNAmmer software (version 1.2) [53,54]. Plastid genes (protein-coding genes [PCGs], tRNAs, and rRNAs) were annotated using an online DOGMA tool with default parameters. A whole mtDNA/cpDNA blast search (E-value ≤ 1 × 10^−5^, minimal alignment length percentage ≥ 40%) was performed against five databases, namely the Kyoto Encyclopedia of Genes and Genomes, Clusters of Orthologous Groups, Non-Redundant Protein, Swiss-Prot, and Gene Ontology [55,56,57,58,59]. The circular map of *S. hemiphyllum* var. *chinense* mtDNA/cpDNA was drawn using OrganellarGenomeDRAW software (version 1.2) [60].

### 4.4. Codon Use Analysis

Using PCGs as the research object, relevant parameters, such as the relative synonymous codon usage (RSCU) and an effective number of codons (ENC) of the tested genes, were analyzed and charted using PhyloSuite software (version 1.2.3) [61], CodonW (version 1.4.2), CUSP biotool software, an online cloud platform (http://cloud.genepioneer.com:9929/#/) (accessed on 11 February 2023), SPSS Statistics (version 17.0), and Microsoft Excel (version 2021). ENC values were plotted against GC3 to generate a standard curve (ENC = 2 + GC3 + 29/[GC3^2^ + (1 − GC3)^2^]). For the R2 plot, G3/(G3 + C3) and A3/(A3 + T3) were used as the *x* and *y* axes, respectively. The central position of the scatter plot represents A=T and G=C. The vector distance between the other points and the center indicates their directional bias. We also performed a neutral point analysis based on GC3 and GC12. To determine the optimal codons, we selected the top and bottom 10% of genes according to ENC values, established high- and low-expression gene pools, then calculated RSCU and ∆RSCU values [∆RSCU = RSCU (high-expression pool) – RSCU (low-expression pool)] to identify the optimal codons with RSCU > 1 and ∆RSCU ≥ 0.08. Cluster analysis was performed to validate the codon preferences.

### 4.5. RNA Editing

We predicted the RNA editing sites of PCGS in the organellar genes of *S. hemiphyllum* var. *chinense* and other species of the three key orders in Phaeophyceae using the Plant Prediction RNA Editor (PREP) tool (http://prep.unl.edu/) (accessed on 9 August 2023) and an online cloud platform (http://cloud.genepioneer.com:9929/#/) (accessed on 11 February 2023).

### 4.6. Repeat Sequence Analysis

We detected the presence of three repetitive sequences (simple, dispersed, and tandem) in the organellar genomes of *S. hemiphyllum* var. *chinense* and other species of the three key Phaeophyceae orders. MIcroSAtellite, a recognition tool Perl script (https://webblast.ipk-gatersleben.de/misa/) (accessed on 26 February 2023), was used to detect simple sequence repeats and identify repeats of one, two, three, four, five, and six nucleotide bases with different repetition numbers. Detection of tandem repeat sequences (>6 bp repeat units) was performed using Tandem Repeats Finder software (version 4.09; http://tandem.bu.edu/trf/trf.submit.options.html) (accessed on 26 February 2023), with default parameters (matching probability of 80 and indel probability of 10). The vmatchPerl script (version 2.3.0) was used to detect the forward, back, reverse, and completion sequences, with the minimum repeat size set to 30 bp.

### 4.7. Comparative Genome and Phylogenetic Analyses

Detailed comparative genomics analysis data of the mtDNA/cpDNA of *S. hemiphyllum* var. *chinense* with those of *S. hemiphyllum* (NC024861/MT800998) and other *Sargassum* species were used to further explore the evolutionary differences in *Sargassum* using Geneious [62] and MEGA software (version 7.0) [63]. Analysis of single nucleotide polymorphism (SNP) sites for 35 PCGs and two open-reading-frames (ORFs) of the mtDNA of *Sargassum* species was performed using DNAMAN software (version 9.0). A comparative genomics study of Fucales, Ectocarpales, and Laminariales was conducted at the organellar genome level to clarify their evolutionary relationship using the progressive Mauve genome aligner (version 2.4.037) with the default settings [64]. A phylogenetic tree was constructed based on 35 shared mtPCGs in *Sargassum* and Phaeophyceae from NCBI with *Coccophora langsdorfii* (NC032287) and *Dictyopteris divaricata* (NC043845) as outgroups. Phylogenetic trees of cpDNA were constructed based on 137 shared PCGs for the genus *Sargassum* with *Coccophora langsdorfii* (NC032287) as an outgroup and 52 shared PCGs for the class Phaeophyceae with *Dictyopteris divaricata* (NC036804) as an outgroup. PCGs were merged into the matrix to select a conserved area (http://phylogeny.lirmm.fr/phylo_cgi/one_task.cgi?task_type=gblocks) (accessed on 9 August 2023). Finally, the phylogenetic tree was constructed using MrBayes software (version 3.2.7a) [65,66]. Phylogenetic analysis was performed using two independent runs with four Markov Chains Monte Carlo simulations for 60,000 (mtPCGs) and 1,000,000 generations (cpPCGs). The output trees were sampled every 100 generations until the average standard deviation of the separation frequencies was <0.01 [67]. The first 25% of the aged samples were removed as burn-in. The final output phylogenetic trees were displayed and edited using FigTree (version 1.3.1; http://tree.bio.ed.ac.uk/) (accessed on 9 August 2023) [68].

### 4.8. Ka/Ks Analysis

We calculated the Ka and Ks values of each shared PCG using the MLWL-based Ka/Ks calculator (version 2.0) and an online cloud platform (http://cloud.genepioneer.com:9929/#/) (accessed on 11 February 2023).

## 5. Conclusions

In this study, we obtained the organellar genomes of *S. hemiphyllum* var. *chinense* with 34,686 bp mitogenome carrying 65 genes and 124,323 bp plastome carrying 173 genes. Comparative genomics analysis showed gene number, type, and arrangement were consistent in organellar genomes of Sargassum, with 360 SNP loci identified as *S. hemiphyllum* var. *chinense* and two genes (*rps*7 and *cox*2) identified as intrageneric classifications of *Sargassum*, the same content and different types (*pet*L was only found in plastomes of the order Fucales and Ectocarpales) and arrangements (most plastomes were rearranged but *trn*A and *trn*D in the mitogenome represented different orders) in genes of the three orders of Phaeophyceae, and most PCGs underwent negative selection (Ka/Ks < 1). Thus far, our research has yielded a comprehensive and reliable phylogenetic tree. However, the absence of nuclear genome information presents challenges in elucidating the phylogenetic relationships within the genus *Sargassum* and among species in the class Phaeophyceae despite high bootstrap and posterior probability values for individual branches based on the existing organellar genome molecular data. In the future, the molecular phylogenetics combining phenotypic variations and nuclear-encoded protein genes will provide additional evidence for the evolutionary distinctions among the class Phaeophyceae species. Furthermore, the inclusion of more whole genome sequences in subsequent research will enhance our understanding of phylogenetic trees and the evolution of brown algae.

## Figures and Tables

**Figure 1 ijms-25-08584-f001:**
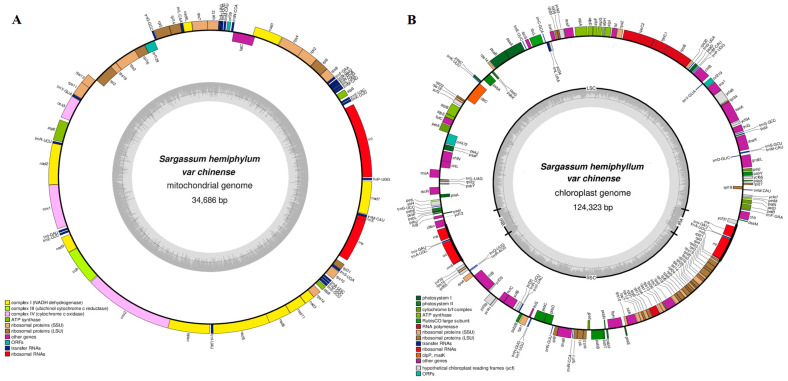
Gene maps of mtDNA (**A**) and cpDNA (**B**) of *Sargassum hemiphyllum* var. *chinense*. The genes on the outside of the maps are transcribed in a clockwise direction, while those on the inside of the maps are transcribed in a counterclockwise direction.

**Figure 2 ijms-25-08584-f002:**
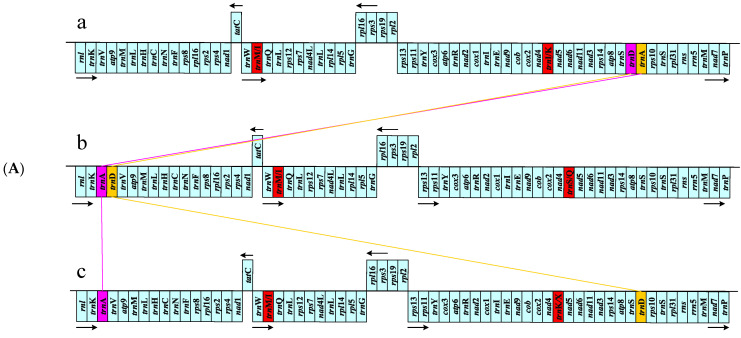
Gene arrangement of (**A**) mtDNAs (**B**) PCGs of cpDNAs, and (**C**) tRNAs and rRNAs of cpDNAs from reported species of three orders: Fucales (**a**), Ectocarpales (**b**), and Laminariales (**c**). Note: The direction of the arrow indicates the orientation of gene coding. Boxes of the same color represent blocks composed of identical genes, while lines of the same color denote the positions of these gene blocks. Genes highlighted in red font signify differential genes among various orders of Phaeophyceae.

**Figure 3 ijms-25-08584-f003:**
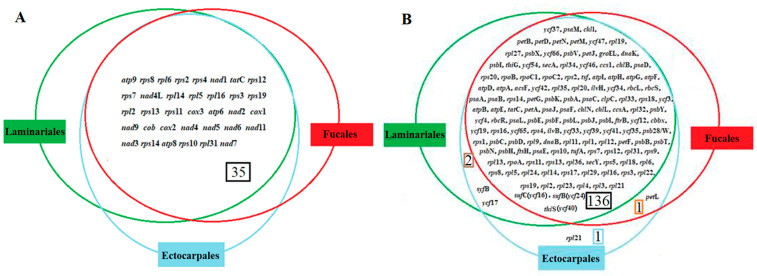
Venn diagram of protein-encoding gene content in (**A**) mtDNAs and (**B**) cpDNAs from reported species of three orders: Fucales, Ectocarpales, and Laminariales. Note: Black Box 35, 35 shared genes of mtDNAs from reported species of three orders: Fucales, Ectocarpales, and Laminariales; black Box 136, 136 shared genes of cpDNAs from reported species of three orders: Fucales, Ectocarpales, and Laminariales; brown Box 2, two shared genes of cpDNAs from reported species of two orders: Ectocarpales and Laminariales; blue Box 1, one unique gene of cpDNAs from reported species of Ectocarpales; orange Box 1, one shared gene of cpDNAs from reported species of two orders: Fucales and Ectocarpales.

**Figure 4 ijms-25-08584-f004:**
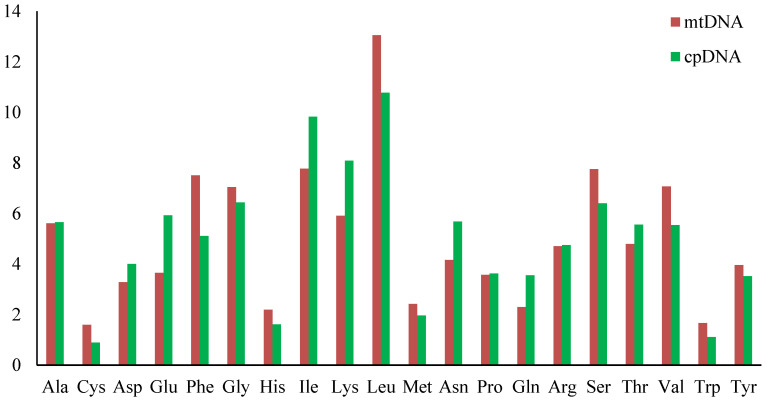
Organellar codon usage pattern of *Sargassum hemiphyllum* var. *chinense*.

**Figure 5 ijms-25-08584-f005:**
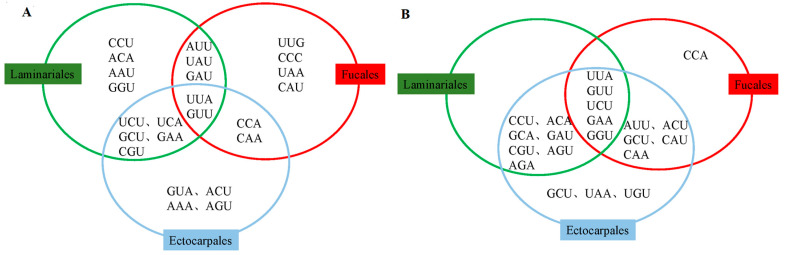
Optimal codons of (**A**) mtDNAs and (**B**) cpDNAs in Fucales, Ectocarpales, and Laminariales.

**Figure 6 ijms-25-08584-f006:**
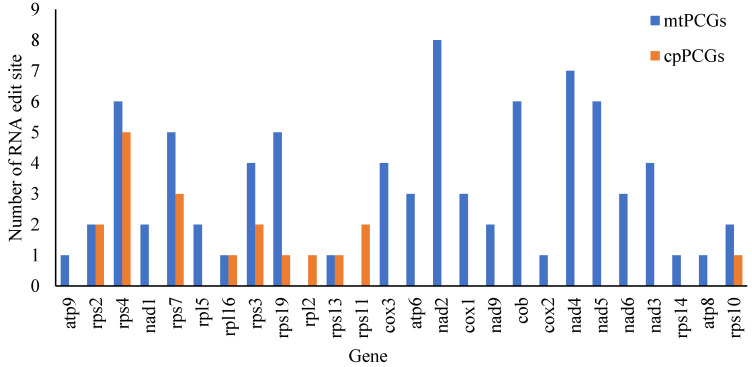
Distribution of RNA-editing sites in the *Sargassum hemiphyllum* var. *chinense* mtPCGs and cpPCGs. The bars represent the number of RNA-editing sites of each gene.

**Figure 7 ijms-25-08584-f007:**
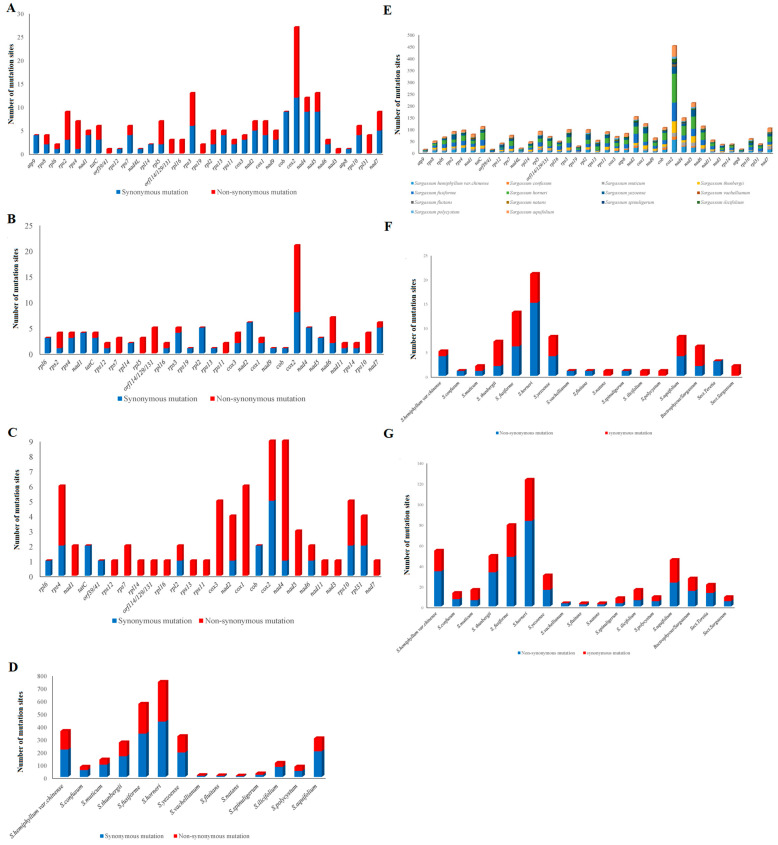
Candidate SNP sites in 14 *Sargassum* species (**A**) subgenera *Bactrophycus* and *Sargassum*, (**B**) grouped Teretia, (**C**) grouped Sargassum, (**D**) SNP analysis of 14 *Sargassum* species, (**E**) 35 PCGs and 2 ORFs, (**F**) *rps*7, (**G**) *cox*2.

**Figure 8 ijms-25-08584-f008:**
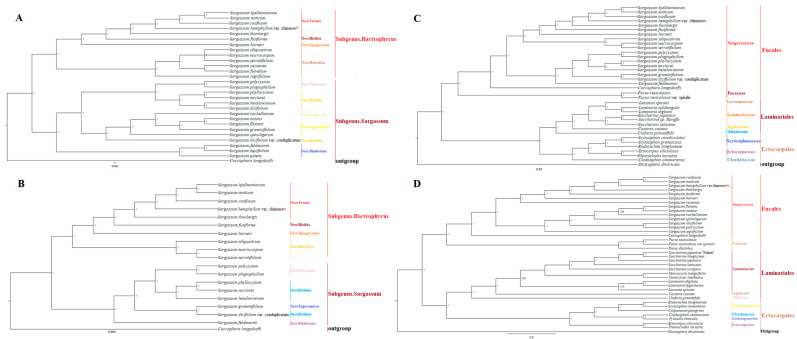
Phylogenetic tree of species related to *Sargassum hemiphyllum* var. *chinense*. The support values for each node were calculated from the Bayesian posterior probability (BPP). The asterisks after the species names indicate newly determined organellar genomes: (**A**) 35 mtPCGs of 28 *Sargassum* species, (**B**) 137 cpPCGs of 18 *Sargassum* species, (**C**) 35 mtPCGs of 38 the Phaeophyceae class species, and (**D**) 52 cpPCGs of 36 the Phaeophyceae class species. Note: (**A**,**B**) represent the phylogenetic tree of species within the genus *Sargassum*, while (**C**,**D**) represent the phylogenetic tree of species within the class Phaeophyceae. The colours of different fonts represent different natural classification positions.

**Figure 9 ijms-25-08584-f009:**
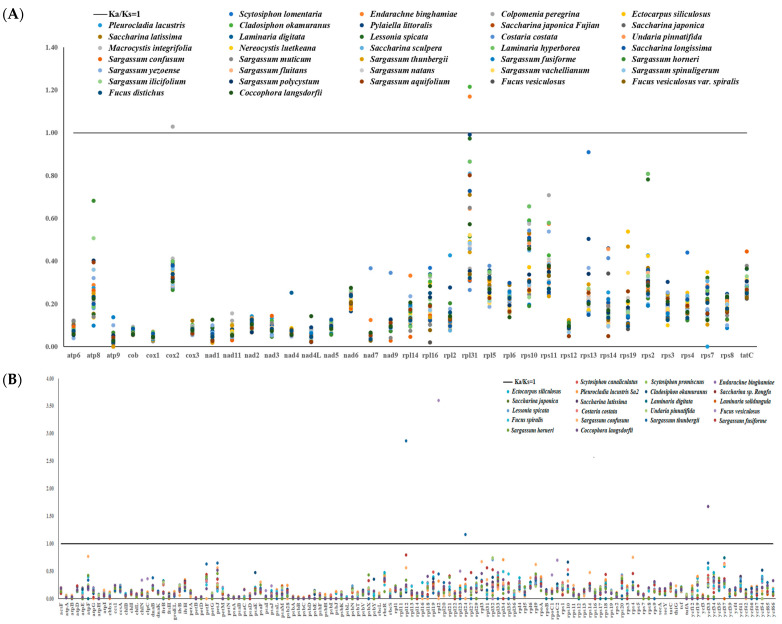
Ka/Ks values for PCGs of Fucales, Ectocarpales, and Laminariales species. (**A**) Analysis was performed for 35 mtPCGs of *Sargassum hemiphyllum* var. *chinense* compared with 36 other species. (**B**) A total of 133 cpPCGs of *Sargassum hemiphyllum* var. *chinense* were analyzed in comparison with 21 other species. Note: The black line in the figure represents the threshold of Ka/Ks = 1. Ka/Ks < 1 indicates purifying selection, Ka/Ks = 1 suggests neutral selection, and Ka/Ks > 1 indicates a positive selection effect.

**Table 1 ijms-25-08584-t001:** Selected species with available mtDNA (*n* = 38) and cpDNA (*n* = 23) in the GenBank. * This study.

Order	Species	Accession No.	Length	GC	PCG	Overlap	Spacer	Genes	PCG	tRNA	rRNA	ORFs	Note
			bp	%	%	bp	bp	Number	Number	Number	Number	Number	
Fucales	*Sargassum hemiphyllum* var. *chinense* *	MT873582	34,686	36.60	75.98	188	1597	65	35	25	3	2	mtDNA
	*Sargassum hemiphyllum* var. *chinense* *	MT873582	124,323	30.60	75.34	75	19,859	173	137	28	6	2	cpDNA
	*Sargassum confusum*	MG459430	34,721	36.60	76.05	180	1596	65	35	25	3	2	mtDNA
	*Sargassum confusum*	MG459429	124,375	30.40	75.30	77	17,439	173	137	28	6	2	cpDNA
	*Sargassum muticum*	NC024614	34,720	36.60	76.01	171	1614	65	35	25	3	2	mtDNA
	*Sargassum thunbergii*	NC026700	34,748	36.60	76.04	168	1600	65	35	25	3	2	mtDNA
	*Sargassum thunbergii*	NC029134	124,592	30.40	75.00	75	17,848	173	137	28	6	2	cpDNA
	*Sargassum fusiforme*	MN883537	34,696	37.50	76.07	211	1570	65	35	25	3	2	mtDNA
	*Sargassum fusiforme*	MN794016	124,298	30.40	75.05	75	17,349	173	137	28	6	2	cpDNA
	*Sargassum horneri*	NC024613	34,606	36.20	76.30	164	1464	65	35	25	3	2	mtDNA
	*Sargassum horneri*	MN265366	124,075	30.60	75.45	75	17,365	173	137	28	6	2	cpDNA
	*Sargassum yezoense*	NC038156	34,767	36.60	75.90	81	1686	65	35	25	3	2	mtDNA
	*Sargassum fluitans*	NC033385	34,727	36.20	75.88	170	1643	65	35	25	3	2	mtDNA
	*Sargassum natans*	NC033384	34,727	36.20	75.90	161	1640	65	35	25	3	2	mtDNA
	*Sargassum vachellianum*	NC027508	34,877	36.20	75.57	158	1787	65	35	25	3	2	mtDNA
	*Sargassum spinuligerum*	NC034272	34,891	36.20	75.41	207	1835	65	35	25	3	2	mtDNA
	*Sargassum ilicifolium*	KT272403	34,925	35.60	75.49	160	1822	65	35	25	3	2	mtDNA
	*Sargassum polycystum*	KT280278	34,862	35.70	75.57	150	1598	65	35	25	3	2	mtDNA
	*Sargassum aquifolium*	NC033408	34,761	36.20	75.89	161	1701	64	35	24	3	2	mtDNA
	*Fucus vesiculosus*	NC007683	36,392	34.40	72.62	121	2086	66	35	25	3	3	mtDNA
	*Fucus vesiculosus*	NC016735	124,986	28.90	74.99	77	17,912	172	137	27	6	2	cpDNA
	*Fucus vesiculosus* var*. spiralis*	MG922856	36,396	34.40	72.69	121	3196	65	35	25	3	2	mtDNA
	*Fucus vesiculosus* var. *spiralis*	MG922855	125,066	28.90	75.02	75	17,919	173	137	28	6	2	cpDNA
	*Fucus distichus*	NC034672	36,400	34.30	72.65	129	2220	64	35	23	3	3	mtDNA
	*Coccophora langsdorfii*	NC032287	35,660	36.40	74.08	194	2512	65	35	25	3	2	mtDNA
	*Coccophora langsdorfii*	NC032288	124,450	28.90	75.10	75	17,846	172	137	27	6	2	cpDNA
Ectocarpales	*Colpomenia peregrina*	NC025302	36,025	32.00	75.30	123	1499	66	35	25	3	3	mtDNA
	*Scytosiphon lomentaria*	NC025240	36,918	34.10	73.40	110	2221	67	35	25	3	4	mtDNA
	*Scytosiphon promiscuus*	MK107985	134,366	31.30	71.91	65	23,496	178	140	28	6	3	cpDNA
	*Endarachne binghamiae*	NC036747	37,460	34.40	72.26	153	2723	67	35	24	3	5	mtDNA
	*Endarachne binghamiae*	NC038231	136,274	31.20	70.80	65	26,270	177	140	28	6	2	cpDNA
	*Ectocarpus siliculosus*	NC030223	37,189	33.50	72.85	199	3153	66	35	25	3	5	mtDNA
	*Ectocarpus siliculosus*	NC013498	139,954	30.70	70.13	65	27,538	185	142	31	6	6	cpDNA
	*Pleurocladia lacustris*	NC032046	37,814	32.90	65.17	226	4134	69	35	24	3	7	mtDNA
	*Pleurocladia lacustris*	NC032045	138,815	29.80	69.62	65	26,578	180	139	30	6	5	cpDNA
	*Cladosiphon okamuranus*	NC040224	38,419	34.30	70.88	128	3628	64	35	24	3	2	mtDNA
	*Cladosiphon okamuranus*	NC046005	137,324	30.20	70.71	65	26,354	178	140	29	6	2	cpDNA
	*Pylaiella littoralis*	NC003055	58,507	38.00	54.35	135	8030	79	36	24	3	16	mtDNA
Laminariales	*Saccharina japonica rongfu*	KX073815	37,638	35.30	72.43	95	2426	66	35	25	3	3	mtDNA
	*Saccharina japonica rongfu*	MK058525	130,584	31.10	73.17	71	21,514	173	138	29	6	2	cpDNA
	*Saccharina japonica*	NC013476	37,657	35.30	72.40	95	2445	66	35	25	3	3	mtDNA
	*Saccharina japonica*	NC018523	130,585	31.10	73.17	71	21,516	173	138	29	6	2	cpDNA
	*Saccharina latissima*	NC026108	37,659	35.40	72.38	95	2519	66	35	24	3	3	mtDNA
	*Saccharina latissima*	NC049039	130,619	31.10	73.15	71	21,638	172	138	28	6	2	cpDNA
	*Saccharina sculpera*	NC029206	37,627	35.20	72.40	95	2624	64	35	24	2	3	mtDNA
	*Saccharina longissima*	NC021640	37,628	35.30	72.39	127	2519	65	35	24	3	3	mtDNA
	*Laminaria solidungula*	NC044690	130,784	31.00	72.67	71	22,118	173	138	29	6	2	cpDNA
	*Laminaria hyperborea*	NC021639	37,976	35.20	71.66	92	2969	64	35	23	3	3	mtDNA
	*Laminaria digitata*	NC004024	38,007	35.10	71.63	148	2405	67	35	25	3	4	mtDNA
	*Laminaria digitata*	NC044689	130,377	31.00	72.69	26	21,920	173	138	29	6	2	cpDNA
	*Lessonia spicata*	NC044181	37,097	32.70	73.21	166	1861	66	35	25	3	3	mtDNA
	*Lessonia spicata*	NC044182	130,305	30.90	73.31	71	21,426	173	138	27	6	2	cpDNA
	*Costaria costata*	NC023506	37,461	34.90	72.76	150	2134	66	35	25	3	3	mtDNA
	*Costaria costata*	NC028502	129,947	30.90	73.50	71	21,064	171	138	27	6	2	cpDNA
	*Undaria pinnatifida*	NC023354	37,402	32.50	72.70	160	2107	65	35	24	3	3	mtDNA
	*Undaria pinnatifida*	NC028503	130,383	30.60	73.24	71	21,449	172	138	28	6	2	cpDNA
	*Macrocystis integrifolia*	NC042669	37,366	32.00	72.93	128	2253	64	35	24	3	2	mtDNA
	*Nereocystis luetkeana*	NC042395	37,399	35.30	72.85	129	2403	63	35	24	2	2	mtDNA
Dictyotales	*Dictyopteris divaricata*	NC043845	32,021	38.30	74.67	160	1370	65	35	24	3	3	mtDNA
	*Dictyopteris divaricata*	NC036804	126,099	31.20	75.64	97	17,191	172	137	27	6	2	cpDNA

**Table 2 ijms-25-08584-t002:** GC content and ENC statistics of codons of Fucales, Ectocarpales, and Laminariales.

	Total Value of Shared CDS (mtDNA)	Total Value of Shared CDS (cpDNA)
GC_all_	35.68% ^a^/32.44% ^b^/34.59% ^c^	31.13% ^a^/31.59% ^b^/31.74% ^c^
GC_1_	41.79% ^a^/42.41% ^b^/43.13% ^c^	41.93% ^a^/42.83% ^b^/42.95% ^c^
GC_2_	36.63% ^a^/36.28% ^b^/37.18% ^c^	34.27% ^a^/34.74% ^b^/34.98% ^c^
GC_3_	28.62% ^a^/18.83% ^b^/23.47% ^c^	17.20% ^a^/17.21% ^b^/17.30% ^c^
ENC	47.12 ^a^/39.41 ^b^/42.77 ^c^	39.44 ^a^/38.87 ^b^/38.28 ^c^

GC_all_, the average GC content of the base at three positions of the codon. GC_1_, GC content of the first base of the codon. GC_2_, GC content of the second base of the codon. GC_3_, GC content of the third base of the code. ^a^, Fucales, ^b^, Ectocarpales, ^c^, Laminariales.

**Table 3 ijms-25-08584-t003:** Prediction of RNA editing sites of *Sargassum hemiphyllum* var. *chinense*.

Type	Effect	^a^ mtPCGs	^b^ cpPCGs	Percentage/%
Hydrophilic	CAT(H)→TAT(Y)	3		^a^ 6.25/^b^ 5.00
Hydrophilic	CAC(H)→TAC(Y)	1	1	
Hydrophilic	CGC(R)→TGC(C)	1		
Hydrophobic	CTC(L)→TTC(F)	1		^a^ 56.25/^b^ 30.00
Hydrophobic	CTT(L)→TTT(F)	12		
Hydrophobic	CGG(R)→TGG(W)	1		
Hydrophobic	CCC(P)→CTC(L)	1		
Hydrophobic	CCT(P)→CTT(L)	8		
Hydrophobic	GCT(A)→GTT(V)	12	3	
Hydrophobic	GCG(A)→GTG(V)	5	1	
Hydrophobic	GCC(A)→GTC(V)	4		
Hydrophobic	GCA(A)→GTA(V)	1	2	
Hydrophilic–hydrophobic	ACG(T)→ATG(M)	1		^a^ 21.25/^b^ 50.00
Hydrophilic–hydrophobic	TCA(S)→TTA(L)	5	3	
Hydrophilic–hydrophobic	TCG(S)→TTG(L)	1		
Hydrophilic–hydrophobic	TCT(S)→TTT(F)	3		
Hydrophilic–hydrophobic	ACA(T)→ATA(I)	2	4	
Hydrophilic–hydrophobic	ACT(T)→ATT(I)	3	3	
Hydrophilic–hydrophobic	ACC(T)→ATC(I)	2		
Hydrophilic–hydrophobic	CCA(P)→TCA(S)	2		^a^ 16.25/^b^ 15.00
Hydrophilic–hydrophobic	CCG(P)→TCG(S)	4		
Hydrophilic–hydrophobic	CCC(P)→TCC(S)	4	1	
Hydrophilic–hydrophobic	CCT(P)→TCT(S)	3	2	

^a^ mtPCGs of *Sargassum hemiphyllum* var. *chinense*, ^b^ cpPCGs of *Sargassum hemiphyllum* var. *chinense*.

## Data Availability

Molecular data have all been deposited to GenBank with the following link: https://www.ncbi.nlm.nih.gov/genbank/ (accessed on 23 December 2020 and 7 November 2021).

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
