# Peer review of "Organellar Genomes of Sargassum hemiphyllum var. chinense Provide Insight into the Characteristics of Phaeophyceae"

_ijms, 2024, doi:10.3390/ijms25168584_

Round 1

Reviewer 1 Report

Comments and Suggestions for Authors

Dear authors,

The manuscript entitled "Organellar Genomes of Sargassum hemiphyllum var. chinense Provide Insight into the Characteristics of Phaeophyceae" sequenced the complete mitochondria and chloroplast genome of S. hemiphyllum var. chinense and investigated the comparative genomics and systematics. It presents scientific relevance for Medicine, Biology, Pharmacy, Chemistry and others area. However, you need to change some details/information in the Abstract, Introduction, Material and Methods, results and discussion, and conclusions.

1. Abstract: Adequate! But:

- The abstract is well written, but there is few information about the methods. Also, I suggest inserting the results obtained (numerical data!!!) more relevant. At the end, I suggest highlighting the advantages/ disadvantages of the study and methods.

2. Introduction section:

- Adequate! I suggest inserting more information about economic and ecological significance of the S. hemiphyllum, and methods used.

- Also, to highlight the advantages/disadvantages/limitations of the study.

3. Results section (or “Results and discussion”)

Wouldn't it be more interesting to combine the "results” with the "discussion" to better describe the findings and compare them with other works published in the literature?? I suggest expanding the discussions!

- Pages 8-9, in “2.4. PCG Codon Use” section, in (lines 229-262): Long paragraph! I suggest dividing it into 2 or 3 paragraphs!

- Page 14, in “2.8. Phylogenetic Analysis and Ka/Ks” section: I suggest improving the quality of Figure 8. The authors wrote "To determine the evolutionary status of Sargassum, a phylogenetic analysis was performed based on 35 PCGs from 27 published mtDNAs and 137 PCGs from 18 published cpDNAs of Sargassum with Coccophora langsdorfii as the outgroup". How was this information previously obtained?

- The results are interesting! I suggest discussing of the results obtained by comparing them with the literature!

4. Discussion section (or “Results and discussion”)?

Wouldn't it be more interesting to combine the "results” with the "discussion" to better describe the findings and compare them with other studies published in the literature?? I suggest expanding the discussions!

- Page 16, in " 3.1. Genomic Characteristics” section (lines 431-468): Long paragraph! I suggest dividing it into 2 or 3 paragraphs! The authors wrote (in lines 461-462) “It is interesting that the arrangement positions of trnA 461 and trnD are unique and representative among the three orders.” I suggest expanding the discussions!

- Page 17, in “3.2. Codon Use Preferences” section: I suggest expanding the discussions and compare them with other studies published in the literature. Same for “3.5. Phylogenetic Analysis and Ka/Ks” section (page 18, lines 540-574).

- I suggest, at the end of the "results and discussion", to write a paragraph summarizing the findings focusing on the advantages/disadvantages/limitations of the study and methods.

5. Materials and methods section: The methodological proposal is appropriate to the manuscript, but I suggest:

- Page 18, in “4.1. Sampling and DNA Extraction” section: What are the conditions for collection/acquisition and storage of samples? What is the time/period (from acquisition to analysis)?

6. Conclusion: There is no 'Conclusions' section??? I suggest inserting the results obtained more relevant. I suggest pointing out the main results and disadvantages/limitations of the method and the study!

7. Table and Figures: Adequate. I suggest improving the quality of all figures!

8. References: Please, check if the references are in accordance with the journal's rules.

Comments on the Quality of English Language

The language (English) is satisfactory (but, I suggest the final revision)! 

Author Response

Dear authors,

The manuscript entitled "Organellar Genomes of Sargassum hemiphyllum var. chinense Provide Insight into the Characteristics of Phaeophyceae" sequenced the complete mitochondria and chloroplast genome of S. hemiphyllum var. chinense and investigated the comparative genomics and systematics. It presents scientific relevance for Medicine, Biology, Pharmacy, Chemistry and others area. However, you need to change some details/information in the Abstract, Introduction, Material and Methods, results and discussion, and conclusions.

Comments 1: Abstract: Adequate! But:- The abstract is well written, but there is few information about the methods. Also, I suggest inserting the results obtained (numerical data!!!) more relevant. At the end, I suggest highlighting the advantages/ disadvantages of the study and methods.

Response 1: Thank you for pointing this out. We agree with this comment. So we modified " We sequenced the complete mitochondria and chloroplast genome of S. hemiphyllum var. chinense and investigated the comparative genomics and systematics." to " Here, we reported the organellar genomes of S. hemiphyllum var. chinense (34,686-bp mitogenome with 65 genes and 124,323-bp plastome with 173 genes) and the investigation of a comparative genomics and systematics of 23 mitogenomes and 38 plastomes of Fucales (including S. hemiphyllum var. chinense), Ectocarpales, and Laminariales in Phaeophyceae." in line 12 of the abstract. We added method "Whole genome collinearity analysis showed" on line 16, result "(nine tandem repeats of 14-24 bp)" on line 26, and result "(Ka/Ks<1)" on line 27. And we revised "Our comprehensive comparative genomic and phylogenetic analyses provide insights into the classification of Sargassum, establishing their respective evolutionary status by elucidating the typical characteristics of three important Phaeophyceae orders." to "Collectively, these findings provide valuable insights to guide future species identification and evolutionary status of three important Phaeophyceae orders species." in line 27 of the abstract.

Comments 2: Introduction section:- Adequate! I suggest inserting more information about economic and ecological significance of the S. hemiphyllum, and methods used.

- Also, to highlight the advantages/disadvantages/limitations of the study.

Response 2: Thank you for pointing this out. We agree with this comment. We modified "Sargassum hemiphyllum is a dominant species in numerous subtropical marine communities, playing a crucial role in the construction of algal field habitats. It contributes to the reduction of eutrophication in aquatic-environments, aids in the repair of damaged habitats, and supports the development of a comprehensive aquaculture model that encompasses multiple trophic levels." to emphasize this point in the 2rd paragraph of the introduction in line 42. And we added "Additionally, Sargassum hemiphyllum possesses significant comer-cial value and can be utilized to extract medicinal and industrial raw materials, including seaweed polysaccharides, alginates, and marine biomass energy, this highlights its potential for economic development and utilization." in the 2rd paragraph of the introduction in line 47.

Comments 3: Results section (or “Results and discussion”) Wouldn't it be more interesting to combine the "results” with the "discussion" to better describe the findings and compare them with other works published in the literature?? I suggest expanding the discussions!

- Pages 8-9, in “2.4. PCG Codon Use” section, in (lines 229-262): Long paragraph! I suggest dividing it into 2 or 3 paragraphs!

- Page 14, in “2.8. Phylogenetic Analysis and Ka/Ks” section: I suggest improving the quality of Figure 8. The authors wrote "To determine the evolutionary status of Sargassum, a phylogenetic analysis was performed based on 35 PCGs from 27 published mtDNAs and 137 PCGs from 18 published cpDNAs of Sargassum with Coccophora langsdorfii as the outgroup". How was this information previously obtained?

- The results are interesting! I suggest discussing of the results obtained by comparing them with the literature!

Response 3: Thank you for reviewing our submitted manuscript and providing valuable feedback. In your suggestions, you mentioned the possibility of merging the results and discussion sections. We appreciate this idea and believe that such a merger could enhance the organization and coherence of the manuscript. However, given the submission guidelines of the journal, we must adhere to the specified structure. Therefore, we will strive to create a closer connection between the results and discussion sections during the revision process to enhance the overall quality and readability of the article.   Thank you for the advice from the reviewer. We agree with this comment. Therefore, we considered breaking down this paragraph into 3 paragraphs in line 249-282 to improve the clarity and logic of the paragraph structure. Thank you again for your valuable feedback.   Thank you very much for pointing this out. We agree with this comment. Therefore, we have optimized and improved the quality of Figure 8 to ensure that the final manuscript meets your requirements.   Thank you for your attention and concern. Our data sources primarily consist of 28 published mtDNAs and 19 published cpDNAs of Sargassum and Coccophora langsdorfii. These data were acquired through NCBI database (https://www.-ncbi.nlm.nih.gov/nuccore/?term=Sargassaceae+complete+genome), which ensures their accuracy and reliability. Among the species of the family Sargassaceae, only the mtDNA and cpDNA of Coccophora langstorfii have been simultaneously published besides those belonging to the genus Sargassum. Consequently, selecting Coccophora langstorfii as an outgroup for phylogenetic analysis may provide a clearer interpretation of the evolutionary position of the genus Sargassum at the mtDNA and cpDNA levels.

Comment 4: Discussion section (or “Results and discussion”)?: -Wouldn't it be more interesting to combine the "results” with the "discussion" to better describe the findings and compare them with other studies published in the literature?? I suggest expanding the discussions!

- Page 16, in " 3.1. Genomic Characteristics” section (lines 431-468): Long paragraph! I suggest dividing it into 2 or 3 paragraphs!

The authors wrote (in lines 461-462) “It is interesting that the arrangement positions of trnA 461 and trnD are unique and representative among the three orders.” I suggest expanding the discussions!

- Page 17, in “3.2. Codon Use Preferences” section: I suggest expanding the discussions and compare them with other studies published in the literature. Same for “3.5. Phylogenetic Analysis and Ka/Ks” section (page 18, lines 540-574).

- I suggest, at the end of the "results and discussion", to write a paragraph summarizing the findings focusing on the advantages/disadvantages/limitations of the study and methods.

Response 4: Thank you for reviewing our submitted manuscript and providing valuable feedback. In your suggestions, you mentioned the possibility of merging the results and discussion sections. We appreciate this idea and believe that such a merger could enhance the organization and coherence of the manuscript. However, given the submission guidelines of the journal, we must adhere to the specified structure. Therefore, we will strive to create a closer connection between the results and discussion sections during the revision process to enhance the overall quality and readability of the article.

Thank you for the advice from the reviewer. We agree with this comment. We considered breaking down this paragraph into 2 paragraphs in 454-498 to improve the clarity and logic of the paragraph structure. Thank you for your valuable feedback.   Thank you very much for pointing this out. We agree with this comment. Therefore, we modified "It is interesting that the arrangement positions of trnA and trnD are unique and representative among the three orders." to "In the mtDNA of the order Fucales and Ectocarpales species, the genes trnD and trnA are closely arranged but their positions of rearrangement differ, the arrangement is trnS-trnD-trnA-rps10 in Fucales species, while in Ectocarpales, it is trnK-trnD-trnA-trnV. In contrast, trnD and trnA are dispersed throughout the mtDNAs of the order Laminariales species, with trnA positioned between trnK and trnV, and trnD located between trnS and rps10. Overall, the trnD-trnA identified in this study represents a rearrangement event of mitochondrial genes within the class Phaeophyceae. This finding is both unique and representative among the three different orders of this class, thereby enhancing our understanding of mitochondrial gene rearrangement in Phaeophyceae." in line 485 in the 2rd paragraph of 3.1. Genomic Characteristics.   Thank you very much for pointing this out. We agree with this comment. We expanded the discussion "The substi-tution rates (Ka/Ks) of 35 mtPCGs and 133 cpPCGs in 21 species of the or-der Fucales, Ectocarpales, and Laminari-ales were almost no non-synymous changes, in-dicating that in most cases, the genes of the class Phaeophyceae species were purified and selected to eliminate harmful mutations while keeps the protein unchanged." in line 584 of page 19 in 3.5. Phylogenetic Analysis and Ka/Ks. We expanded the discussion "There is one thing worth paying attention to Sargassum siliquastrum, Sargassum macrocarpum, and Sargassum serrati-folium are observed on the same branch and node with high support rates in Figure 8A. However, in the phyloge-netic trees depicted in Figures 8B and 8C, these species are located on diffe-rent nodes of the same branch with high support rates. It is speculated that analytical factors (such as insufficient number of genes or loci) and biological factors (such as incomplete lineage sorting) play a certain role in the evoluti-on of Sargassum siliquastrum, Sargassum macrocarpum, and Sargassum serratifolium.This result provides a summary of the actual classification and the id-entification key for the subdivision of Sargassum." in line 605 of page 20 in 3.5. Phylogenetic Analysis and Ka/Ks. We expanded the discussion "The conservation within the Sargassum genus is supported by phylogenetic analysis, which revealed that all members of Sargassum are grouped into a single clade (Fucales) according to their taxonomic order. This con-sistent clus-tering suggests that there has been no rapid evolution of the Sargassum genus concerning organellar genomes. Notably, S. kjellmannianum and S. muticum exhibited extremely high sequence similarity and were always identified as sister species phylogenetically in both chloroplast and mitochondrial genomes. However, " in line 614 of page 20 in 3.5. Phylogenetic Analysis and Ka/Ks. We expanded the discussion "165 PCGs (132 chloroplast PCGs and 32 mitochondrial PCGs) shared in 50 species" in line 626 of page 20 in 3.5. Phylogenetic Analysis and Ka/Ks. We expanded the discussion ", but we did not consider their morphological characteristics or other genetic factors" in line 628 of page 20 in 3.5. Phylogenetic Analysis and Ka/Ks. We expanded the discussion "The species in the order Fucales, Laminariales, and Ectocarpales form independent branches, and the first two form a large branch, this means that" in line 630 of page 20 in 3.5. Phylogenetic Analysis and Ka/Ks.   Thank you very much for pointing this out. We agree with this comment. Therefore, we wrote a paragraph "5. Conclusion In this study, we obtained the organellar genomes of S. hemiphyllum var. chinense with 34,686-bp mitogenome carrying 65 genes and 124,323-bp plastome carrying 173 genes. Comparative genomics analysis showsed gene number, type, and arrangement were consistent in organellar genomes of Sargassum with 360 SNP loci identified as S. hemiphyllum var. chinense and two genes (rps7 and cox2) identified as intrageneric classifications of Sargassum, the same content and different types (petL was only found in plastomes of the order Fucales and Ectocarpales) and arrangements (most plastomes were rearranged but trnA and trnD in the mitogenome represented different orders) in genes of the three orders of Phaeophyceae, and most PCGs underwent negative selection (Ka/Ks<1). Thus far, our research has yielded a comprehensive and reliable phylogenetic tree. However, the absence of nuclear genome information presents challenges in elucidating the phylogenetic relationships within the genus Sargassum and among species in the class Phaeophyceae, despite high bootstrap and posterior probability values for individual branches based on the existing organellar genome molecular data. In the future, the molecular phylogenetics combining phenotypic variations and nuclear-encoded protein genes will provide additional evidence for the evolutionary distinctions among the class Phaeophyceae species. Furthermore, the inclusion of more whole genome sequences in subsequent research will enhance our understanding of phylogenetic trees and the evolution of brown algae." in line 637 of page 20.

Comment 5: Materials and methods section: The methodological proposal is appropriate to the manuscript, but I suggest:

- Page 18, in “4.1. Sampling and DNA Extraction” section: What are the conditions for collection/-acquisition and storage of samples? What is the time/period (from acquisition to analysis)?

Response 5: Thank you very much for pointing this out. We agree with this comment. Therefore, we modified "Subsequently, the algae were transported to the laboratory in an insulated box, maintain-ing a temperature range of 20℃ to 26℃. They were then cultured in sterile filtered seawater for 3 d under a fluorescent light (80-110 μmol photons m–2 s–1; 12 h light/dark cycles) at 24°C-26°C in laboratory." in line 661 and "and being dried with paper towels" in line 666 of page 21 in 4.1. Sampling and DNA Extraction section.

Comment 6: Conclusion: There is no 'Conclusions' section??? I suggest inserting the results obtained more relevant. I suggest pointing out the main results and disadvantages/limitations of the method and the study!

Response 6: Thank you very much for pointing this out. We agree with this comment. Therefore, we wrote a paragraph "5. Conclusion In this study, we obtained the organellar genomes of S. hemiphyllum var. chinense with 34,686-bp mitogenome carrying 65 genes and 124,323-bp plastome carrying 173 genes. Comparative genomics analysis showsed gene number, type, and arrangement were consistent in organellar genomes of Sargassum with 360 SNP loci identified as S. hemiphyllum var. chinense and two genes (rps7 and cox2) identified as intrageneric classifications of Sargassum, the same content and different types (petL was only found in plastomes of the order Fucales and Ectocarpales) and arrangements (most plastomes were rearranged but trnA and trnD in the mitogenome represented different orders) in genes of the three orders of Phaeophyceae, and most PCGs underwent negative selection (Ka/Ks<1). Thus far, our research has yielded a comprehensive and reliable phylogenetic tree. However, the absence of nuclear genome information presents challenges in elucidating the phylogenetic relationships within the genus Sargassum and among species in the class Phaeophyceae, despite high bootstrap and posterior probability values for individual branches based on the existing organellar genome molecular data. In the future, the molecular phylogenetics combining phenotypic variations and nuclear-encoded protein genes will provide additional evidence for the evolutionary distinctions among the class Phaeophyceae species. Furthermore, the inclusion of more whole genome sequences in subsequent research will enhance our understanding of phylogenetic trees and the evolution of brown algae." in line 637 of page 20.

Comment 7: Table and Figures: Adequate. I suggest improving the quality of all figures!

Response 7: Thank you for pointing this out. We agree with this comment. Therefore, we adjusted the quality of all figures as much as possible. Please refer to the revised manuscript for details.

Comment 8: References: Please, check if the references are in accordance with the journal's rules.

Response 8: Thank you for pointing this out. We agree with this comment. Therefore, we checked the reference format again to ensure it conforms to the journal's rules. Please refer to the references of revised manuscript for details.

Reviewer 2 Report

Comments and Suggestions for Authors

Manuscript ID:  ijms-3105530

Authors: Jia et al.

In this research article entitled “Organellar Genomes of Sargassum hemiphyllum var. chinense Provide Insight into the Characteristics of Phaeophyceae”, the authors provided a comparative study of structure and evolutionary characteristics of the organelle genomes, coding gene arrangement, structure–function relationships, and systematic evolution of Sargassum hemiphyllum var. chinense. They also focused on the codon preference, RNA editing sites, and repetitive sequences for the three major orders of Phaeophyceae: Fucales, Ectocarpales, and Laminariales.

Hereafter, some points that should be taken into account before processing further.

Comments to the authors:

1-      It would be better to merge the cells of table 1 that have the same order.

2-      The authors should make some of the annotations clearer to the reader. The font size should be increased such as in figures 3, 8 and 9.

3-      The presence of several nodes on the same branch is overlooked.

4-      Additional value will be added if the authors add the limitations of the study at the last part of the discussion.

5-      The English language is fine but small editing is required.

Comments on the Quality of English Language

The English language of the manuscript is fine just minor editing is required.

Author Response

In this research article entitled “Organellar Genomes of Sargassum hemiphyllum var. chinense Provide Insight into the Characteristics of Phaeophyceae”, the authors provided a comparative study of structure and evolutionary characteristics of the organelle genomes, coding gene arrangement, structure–function relationships, and systematic evolution of Sargassum hemiphyllum var. chinense. They also focused on the codon preference, RNA editing sites, and repetitive sequences for the three major orders of Phaeophyceae: Fucales, Ectocarpales, and Laminariales.

Hereafter, some points that should be taken into account before processing further. Comments to the authors:

Comments 1: It would be better to merge the cells of table 1 that have the same order.

Response 1: Thank you for pointing this out. We agree with this comment. We have, accordingly, changed the format of table1 to better display its contents in line 113 in page 3-6.

Comments 2:-The authors should make some of the annotations clearer to the reader. The font size should be increased such as in figures 3, 8 and 9.

Response 2: Thank you for pointing this out. We agree with this comment. We have, accordingly, done some of the annotations "Note: Black Box 35, 35 shared genes of mtDNAs from reported species of three orders: Fucales, Ectocarpales, and Laminariales; Black Box 136, 136 shared genes of cpDNAs from reported species of three orders: Fucales, Ectocarpales, and Laminariales; Brown Box 2, 2 shared genes of cpDNAs from reported species of two orders: Ectocarpales and Laminariales; Blue Box1, 1 unique gene of cpDNAs from reported species of Ectocarpales; Orange Box 1, 1 shared gene of cpDNAs from re-ported species of two orders: Fucales and Ectocarpales." for better understanding by readers in line 228 and increased the font size in Figure 3.

We have, accordingly, done some of the annotations "Note: A and B represent the phylogenetic tree of species within the genus Sargassum, while C and D represent the phylogenetic tree of species within the class Phaeophyceae." for better understanding by readers in line 410 in Figure 8. We have, accordingly, done some of the annotations "Note: The black line in the figure represents the threshold of Ka/Ks = 1. Ka/Ks < 1 indicates purifying selection, Ka/Ks = 1 suggests neutral selection, and Ka/Ks > 1 indicates a positive selection effect." for better understanding by readers in line 451 in Figure 9.

Comments 3: -The presence of several nodes on the same branch is overlooked.

Response 3: Thank you for pointing this out. In response to this point, we provide the following explanation: In Figure 8A, Sargassum siliquastrum, Sargassum macrocarpum, and Sargassum serratifolium are observed on the same branch and node with high support rates. However, in the phylogenetic trees depicted in Figures 8B and 8C, these species are located on different nodes of the same branch with high support rates. Despite the continuous advance-ments and cost reductions in genome sequencing technology, there is an increasing trend in research to utilize large-scale omics data-specifically phylogenetic genomics which significantly aids in clarifying the important branches of the tree of life. Nevertheless, the phenomenon of inconsistent or even conflicting evolutionary trees remains prevalent. For these confli-cting evolutionary trees, even when employing whole genome data and the most advanced algorithms, clear answ-ers are still elusive. Studies indicate that the reasons for the inconsistency among common evolutionary trees can be broadly categorized into two grou-ps: analytical factors and biological factors. Furthermore, defects in the software algorithms used to construct evolutionary trees can also contribute to the discrepancies observed. Given the universality of these inconsistencies, we aim to incorporate the results of all evolutionary trees as comprehen-sively as possible; thus, we select 'representative' trees to elaborate on the main conclusions.

Comments 4: -Additional value will be added if the authors add the limitations of the study at the last part of the discussion.

Response 4: Thank you for pointing this out. We agree with this comment. We have, accordingly, added "Thus far, our research has yielded a comprehensive and reliable phylogenetic tree. However, the absence of nuclear genome information presents challenges in elucidating the phylogenetic relationships within the genus Sargassum and among species in the class Phaeophyceae, despite high bootstrap and posterior probability values for individual branches based on the existing organellar genome molecular data. In the future, the molecular phylogenetics combining phenotypic variations and nuclear-encoded protein genes will provide additional evidence for the evolutionary distinctions among the class Phaeophyceae species. Furthermore, the inclusion of more whole genome sequences in subsequent research will enhance our understanding of phylogenetic trees and the evolution of brown algae." in line 646 of conclusion to emphasize this point.

Comments 5: -The English language is fine but small editing is required.

Response 5: Thank you for your comments and suggestions on our manuscript. We appreciate your feedback and would like to address the issue you raised the minor editing in the English language. Our manuscript has already been edited by a professional editing service - editage (https://www.editage.cn/info/ppc/brand_I?utm_source=baidu&utm_medium=cpc&utm_camp-aign=brand&utm_term=%E6%84%8F%E5%BE%97%E8%BE%91&sdclkid=AL2z15fD15fNASDixOF&bd_vid=8902204016825827760). However, we still taked your feedback and carefully reviewed the manuscript again to ensure that any necessary corrections are made to improve the clarity and accuracy of the language. Thank you for your valuable input and we look forward to submitting the revised manuscript for your review.

Reviewer 3 Report

Comments and Suggestions for Authors

This manuscript presents a comprehensive study on the organellar genomes of Sargassum hemiphyllum var. chinense, offering valuable insights into the systematics and phylogenetics of the Phaeophyceae orders Fucales, Ectocarpales, and Laminariales. The research is significant due to the economic and ecological importance of this seaweed and the ongoing debates about its classification.

Overall, I found manuscript is very good in shape. The introduction provides necessary background information and sets the context for the study. The objectives of the study could be more explicitly stated to guide the reader. The methods section is thorough, describing the sequencing and analysis processes. The criteria for identifying SNP loci and RNA-editing sites should be explicitly stated. The results are well-presented, with significant findings on gene number, type, and arrangement in the organellar genomes. The identification of 360 SNP loci and the specific genes associated with intrageneric classifications of Sargassum are particularly noteworthy. The comparative genomics analysis across the three Phaeophyceae orders is detailed and informative. The discussion interprets the results well, providing insights into Sargassum classification and the evolutionary relationships among Phaeophyceae orders.

Author Response

Comments and Suggestions for Authors

Comments 1: This manuscript presents a comprehensive study on the organellar genomes of Sargassum hemiphyllum var. chinense, offering valuable insights into the systematics and phylogenetics of the Phaeophyceae orders Fucales, Ectocarpales, and Laminariales. The research is significant due to the economic and ecological importance of this seaweed and the ongoing debates about its classification.

Overall, I found manuscript is very good in shape. The introduction provides necessary background information and sets the context for the study. The objectives of the study could be more explicitly stated to guide the reader. The methods section is thorough, describing the sequencing and analysis processes. The criteria for identifying SNP loci and RNA-editing sites should be explicitly stated. The results are well-presented, with significant findings on gene number, type, and arrangement in the organellar genomes. The identification of 360 SNP loci and the specific genes associated with intrageneric classifications of Sargassum are particularly noteworthy. The comparative genomics analysis across the three Phaeophyceae orders is detailed and informative. The discussion interprets the results well, providing insights into Sargassum classification and the evolutionary relationships among Phaeophyceae orders.

Response 1: We appreciate your positive feedback on our manuscript and are pleased to learn that you found the study valuable and impactful. We have thoughtfully considered your suggestions and made the necessary revisions to enhance the quality of the manuscript. Thank you again for your time and effort in reviewing our work.

Round 2

Reviewer 1 Report

Comments and Suggestions for Authors

The authors improve the quality of the manuscript based on my observations.

Comments on the Quality of English Language

Only, Final Revision.